# MULTI-GRANULARITY SEMANTIC REVISION FOR LARGE LANGUAGE MODEL DISTILLATION

## ABSTRACT

Knowledge distillation plays a key role in compressing the Large Language Models (LLMs), which boosts a small-size student model under large teacher models' guidance. However, existing LLM distillation methods overly rely on student-generated outputs, which may introduce generation errors and misguide the distillation process. Moreover, the distillation loss functions introduced in previous works struggle to align the most informative part due to the complex distribution of LLMs' outputs. To address these problems, we propose a multi-granularity semantic revision method for LLM distillation. At the sequence level, we propose a sequence correction and re-generation (SCRG) strategy. SCRG first calculates the semantic cognitive difference between the teacher and student to detect the error token, then corrects it with the teacher-generated one, and re-generates the sequence to reduce generation errors. At the token level, we design a distribution adaptive clipping Kullback-Leibler (DAC-KL) loss as the distillation objective function. DAC-KL loss exploits a learnable sub-network to adaptively extract semantically dense areas from the teacher's output, avoiding the interference of redundant information in the distillation process. Finally, at the span level, we leverage the span priors of a sequence to compute the probability correlations within spans, and constrain the teacher and student's probability correlations to be consistent, further enhancing the transfer of semantic information. Extensive experiments across different model families with parameters ranging from 0.1B to 13B demonstrate the superiority of our method compared to existing methods.

## 1 INTRODUCTION

The remarkable advancements in auto-regressive Large Language Models (LLMs) (Kaplan et al., 2020; Wei et al., 2022; Radford et al.; Zhang et al.; Brown et al., 2020) have led to unprecedented breakthroughs in a diverse array of text generative tasks, with numerous open-source models (Touvron et al.; Zhang et al., 2022) now available. A crucial factor contributing to this success is the ability to scale up the models, which involves increasing both the amount of training data and the number of model parameters. However, the massive size and computational intensity of these state-of-the-art models pose significant challenges, particularly when it comes to deployment and real-time applications. In contrast, smaller models with limited parameters often sacrifice performance on real-world generation tasks (Wang et al., 2022a). To mitigate these challenges, Knowledge Distillation (KD) (Hinton et al., 2015) has emerged as a pivotal technique, enabling the development of smaller, more efficient student models that inherit the strengths of their larger teacher counterparts.

Traditional knowledge distillation methods (Hinton et al., 2015; Kim & Rush, 2016) directly employ Kullback-Leibler divergence (KLD) as the distillation loss for aligning the output distributions of teacher and student models on a static dataset (see Figure 1 (a)). Unlike these methods, recent LLM distillation methods are exploring diverse divergence loss functions tailored to LLMs and leveraging student-generated datasets to avoid distribution mismatch between the outputs student-generated in the training and inference stages. GKD (Agarwal et al., 2024) and MiniLLM (Gu et al., 2023) propose to exploit reverse KLD as the distillation objective, replacing the commonly used forward KLD. These approaches aim to prevent students from overestimating the low-probability regions of the teacher's distribution. Also, these methods train the student on self-generated sequences that are on-policy instead of a fixed set of output sequences. Recently, Distillm (Ko et al., 2024) proposes an adaptive off-policy student-generation strategy to improve the sample efficiency and high generation

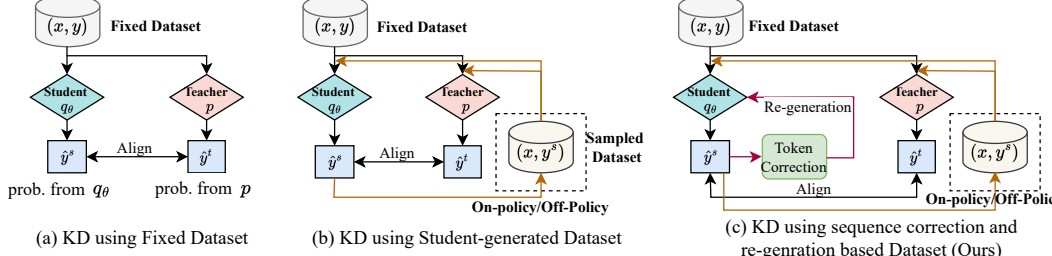

Figure 1: Knowledge Distillation using Different Sampled Datasets. (a) Traditional KD using a fixed dataset (Hinton et al., 2015). (b) KD using the student-generated dataset, which can be categorized into on-policy based methods (Agarwal et al., 2024; Gu et al., 2023) and the off-policy based method (Ko et al., 2024). (c) Our proposed KD approach, which leverages a sequence correction and re-generation strategy and can be seamlessly integrated with both on-policy and off-policy generation schedules.

time faced in on-policy generation (see Figure 1 (b)). Meanwhile, it designs a new distillation object function *i.e.*, skew KLD loss for better generalizability and convergence. However, relying on student-generated sequences may introduce generation errors and lead to suboptimal learning, as the distillation process becomes vulnerable to the inaccuracies inherent in the student's predictions. The student model's limited capacity and biases can further perpetuate these errors, resulting in a distorted representation of the teacher's knowledge. Moreover, the rich semantic knowledge and the significant variance across different tokens make it challenging for existing distillation objective functions to capture and transfer the essential knowledge within the teacher model's output distribution.

To address the above-mentioned issues, we introduce a novel multi-level semantic revision approach, across sequence token and span levels, to significantly improve the KD process for LLMs. At the sequence level, we propose a sequence correction and re-generation (SCRG) strategy. We detect the error token in the student-generated sequence and re-generate the sequence from the position of the error token to reduce generation errors. As shown in Figure 1 (c), by assessing the semantic cognitive differences between teacher and student outputs on a token-by-token basis, we identify and correct errors, leading to re-generated sequences that steer the student model towards generating more reliable samples and can be seamlessly integrated with both on-policy and off-policy generation schedules. At the token level, we employ a distribution adaptive clipping Kullback-Leibler (DAC-KL) loss function, which leverages a learnable sub-network to target semantically salient regions of the output distribution. This loss function effectively filters out redundant information, preserving only the most relevant signals for distillation. Finally, at the span level, we incorporate pre-defined span priors of sequences to align the relations of probability vectors of the student and teacher models, ensuring a consistent transfer of semantic information across related tokens within the same span. Through extensive experiments with different models, including the LLAMA2, OpenLLAMA2, OPT, and GPT2 series, ranging from 0.1B to 13B parameters, we showcase the superiority of our approach over existing knowledge distillation methods.

The contributions of this paper are summarized as follows:

- We introduce a novel multi-level semantic revision approach to enhance the knowledge distillation (KD) process for large language models (LLMs).

- At the sequence level, we propose a sequence correction and re-generation strategy to steer the student model towards generating more reliable sequences.

- At the token level, we propose a distribution adaptive clipping Kullback-Leibler loss to capture semantically salient regions of the output space.

- At the span level, we incorporate input span priors to ensure a consistent transfer of semantic knowledge across related tokens.

- Through extensive experimentation with models ranging from 0.1B to 13B parameters, we demonstrate the superiority of our method over existing KD methods for LLMs.

## 2 RELATED WORK

**KD for encoder-only language models.** Pretrained encoder-only language models, such as BERT (Jiao et al., 2019), can be compressed using the traditional logit distillation (Hinton et al., 2015) and feature distillation (Adriana et al., 2015). These knowledge distillation methods minimize the Kullback-Leibler divergence loss between the outputs of the student and teacher models on a fixed dataset (Kim & Rush, 2016). Liang et al.(Liang et al., 2020) applied this objective to train students on masked language modelling and text classification tasks. Jiao et al.(Jiao et al., 2019) utilized intermediate representations from each transformer layer of the teacher as transferable knowledge. Despite the potential of KD in encoder-only language models (Sanh et al., 2019; Liang et al., 2023; Sun et al., 2019; Liu et al., 2022), the complex predictions generated by large language models (LLMs) through auto-regressive inference present a new challenge. This paper primarily discusses KD for auto-regressive LLMs.

**KD for auto-regression large language models.** Existing knowledge distillation (KD) methods for auto-regressive large language models (LLMs) can be divided into black-box methods for closed-source models such as GPT-3.5 (Ouyang et al., 2022) and GPT-4 (Achiam et al., 2023), and white-box methods for open-source models such as LLaMA (Touvron et al.). Black-box methods (Chen et al., 2024; Jiang et al., 2023; Hsieh et al., 2023) cannot access the internal parameters of the teacher model and utilize only the inference results provided by the teacher API (Taori et al., 2023; Chiang et al., 2023; Peng et al., 2023). The inference results of the teacher model are used to construct prompt-response pairs, which serve as a new training dataset to fine-tune the student model. In contrast, white-box KD methods (Ko et al., 2024; Agarwal et al., 2024; Gu et al., 2023) leverage the internal parameters of the teacher model, providing richer training signals such as the probability distribution of predictions, potentially leading to better student model performance. Our methods primarily address the challenges of existing methods in the realm of white-box KD.

## 3 PRELIMINARY

Before introducing our method, we provide some preliminary information on KD for LLMs. We consider the inference of LLMs as a vocabulary classification task, where a model $p$ predicts the conditional probability distribution of a target response $y$ given a prompt and target sequence pair $(x, y)$. Let $y_{<i} = (y_1, y_2, ...., y_{i-1})$ denote the generated output sequence up to the $(i-1)^{th}$ token $y_{i-1}$. A token-level auto-regression model outputs a next-token $M-$vocabulary probability distribution. Specifically, for the model $p$, $\hat{y}_i = p(.|y_{<i}, X)(\hat{y}_i \in \mathbb{R}^M)$ represents the probability distribution of the generated $i^{th}$ token, where $\hat{y}_i \in (0, 1)^M$. $y_i \sim p(.|y_{<i}, X)$ is the corresponding output token.

We formulate KD as an optimization problem that aims to minimize the difference between the prediction distribution of a fixed teacher model $p(.|y_{<i}, x)$ and that of a parameterized student model $q_\theta(.|y_{<i}, x)$, using sampled input-output sequence pairs $(x,y)$ from the fixed dataset $(X,Y)$. $\theta$ is the student's parameters to be optimized. The sequence-level distillation with $L_y$ tokens employs KL Divergence $D_{KL}$ as the distillation object. The total distillation loss $\mathcal{L}_{KD}$ is broken down into a sum of token-wise distillation:

$$\mathcal{L}_{KD} = \frac{1}{L_y} \sum_{i=1}^{L_y} D_{KL}(p(.|y_{<i}, x)||q_\theta(.|y_{<i}, x)) = \frac{1}{L_y} \sum_{i=1}^{L_y} p(.|y_{<i}, x) log \frac{p(.|y_{<i}, x)}{q_\theta(.|y_{<i}, x))}, \quad (1)$$

where the conditional sequence $y$ can be easily generated by sampling from the teacher or student model policy, *i.e.*,$\{x \in X, y \sim p(.|x)\}$ or $\{x \in X, y \sim q_\theta(.|x)\}$ instead of directly $\{(x, y) \in (X, Y)\}$.

During the distillation process, the student model is also guided by the ground-truth output sequence without querying the policies of the teacher or student models. The supervised fine-tuning (SFT) loss is formulated as

$$\mathcal{L}_{\text{SFT}} = \mathbb{E}_{(x,y)\sim(X,Y)}[-log\, q_\theta(y|x)]. \quad (2)$$

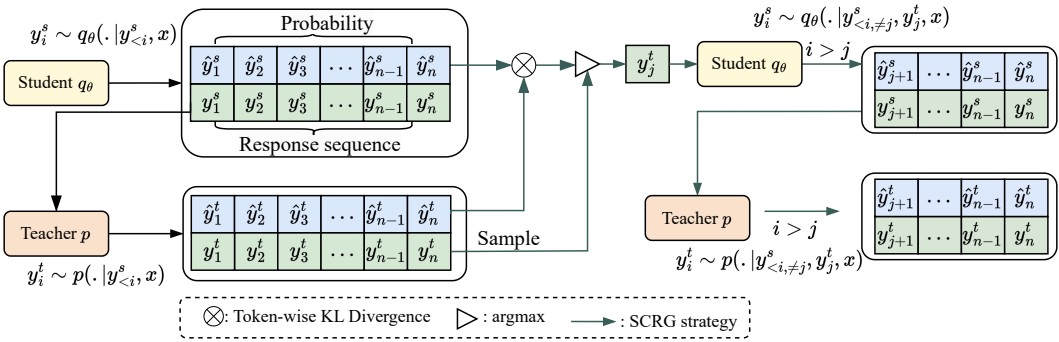

Figure 2: The workflow of sequence correction and re-generation strategy.

# 4 MULTI-GRANULARITY SEMANTIC REVISION

In this section, we introduce the proposed multi-granularity semantic revision for LLM distillation, which revises the semantic representation during the knowledge transfer stage at three levels: sequence-level, token-level, and span-level.

## 4.1 SEQUENCE-LEVEL CORRECTION AND RE-GENERATION

As illustrated by Eq. equation 1, prevalent KD methods (Agarwal et al., 2024; Gu et al., 2023; Ko et al., 2024), utilizes conditional sequences generated from the student model (denoted as $y \sim q_\theta(\cdot|x)$ ) for the distillation process. While these methods are designed to mitigate the training-inference mismatch between the fixed training data and the student's auto-regressive inferences, they simultaneously risk introducing generation errors. Due to the limited capabilities of the student model, the generated sequences may contain additional errors which reduces the effectiveness of KD. To address this issue, we propose a sequence correction and re-generation (SCRG) strategy (shown in Fig. 2) to detect generation errors and re-generate sequences that steer the student model towards generating reliable sequences.

We denote the generated $n$-token sequence from the student model $q_\theta$ as $y^s_{<n+1} = (y^s_1, y^s_2, ...., y^s_n)$ which correspondences the probability outputs $(\hat{y}^s_1, \hat{y}^s_2, ...., \hat{y}^s_n)$, where $y^s_i \sim q_\theta(\cdot|y^s_{<i}, x)(1 \le i \le n)$. Similarly, we denote the teacher model's output sequence as $y^t_{<n+1} = (y^t_1, y^t_2, ...., y^t_n)$ and probability outputs $(\hat{y}^t_1, \hat{y}^t_2, ...., \hat{y}^t_n)$. We denote each token of the teacher model's output sequence as $y^t_i \sim p(\cdot|y^s_{<i}, x)$. We follow previous methods (Agarwal et al., 2024; Gu et al., 2023; Ko et al., 2024) using the student-generated outputs as the distillation dataset, and calculate token-wise KLD loss to evaluate the semantic cognitive differences between the teacher and student for each token to detect the position of the error token within the sequence $y^t_{<n+1}$. We formulate the detection process of the error token $y^s_j$ as

$$j = \operatorname*{arg\,max}_{1 \le i \le n} \left( D_{KL}(\hat{y}^s_i \| \hat{y}^t_i) \text{ if } y^s_i \ne y^t_i \right). \tag{3}$$

We then replace the $y^s_j$ by $y^t_j$ to construct new samples and re-generate the student output sequence and each token in $y^s_{<n+1}$ is formulated as

$$y^s_i \sim \begin{cases} q_\theta(\cdot|y^s_{<i}, x) & \text{if } i < j \\ p(\cdot|y^s_{<i}, x) & \text{if } i = j \\ q_\theta(\cdot|y^s_{<i, \ne j}, y^t_j, x) & \text{if } i > j. \end{cases} \tag{4}$$

Our SCRG strategy can be seamlessly integrated with existing on-policy sampling (Agarwal et al., 2024) and off-policy sampling (Ko et al., 2024). By incorporating an adaptive scheduler (Ko et al., 2024) for student-model generation, we enhance the efficiency of our sampling process.

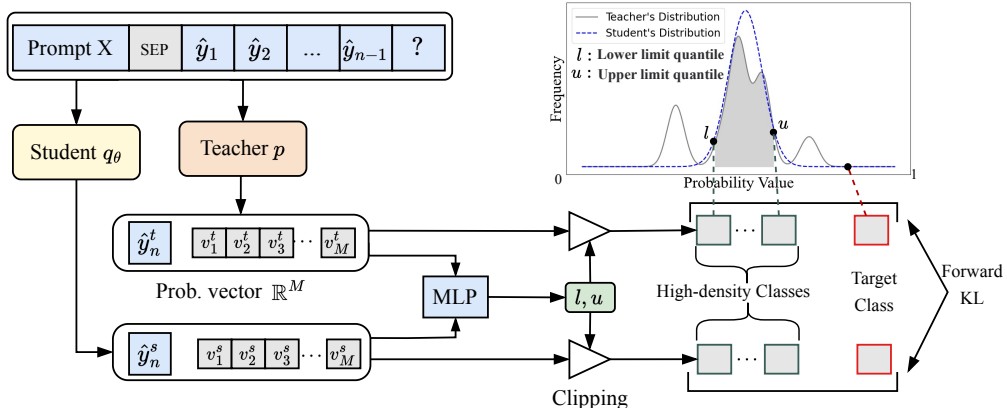

Figure 3: The workflow of the DAC-KL loss function.

## 4.2 TOKEN-LEVEL DAC-KL LOSS FUNCTION

The probability output of LLMs is a high-dimensional vector for each token. However, existing modified Kullback-Leibler divergence (KLD) loss functions, used as knowledge distillation objectives, struggle to effectively capture the valuable distribution with high semantic knowledge from the teacher network. They either underfit the teacher's distribution, as seen in forward KLD, or tend to overfit to a part of the high-probability region, as seen in reverse KLD. To address this issue, we design a Distribution-Adaptive Clipping Kullback-Leibler (DAC-KL) loss function (in Fig. 3) to capture high-density semantic regions of the teacher's output probability distribution, which can be more easily imitated by the student models with limited capacity.

The probability outputs at the $i^{th}$ token position of both the teacher and student models are high-dimensional probability vectors with $M$ tokens, which are denoted as

$$
\begin{aligned}
\hat{y}_i^t &= p(.|y_{<i}^s, x) = [v_1^t, v_2^t, ..., v_M^t] \in \mathbb{R}^M, \\
\hat{y}_i^s &= q_\theta(.|y_{<i}^s, x) = [v_1^s, v_2^s, ..., v_M^s] \in \mathbb{R}^M.
\end{aligned}
\tag{5}
$$

We input these two probability vectors to a learnable MLP sub-network $f_{sub}$ to predict the upper limit quantile $u \in [0, 1]$ and the lower limit quantile $l \in [0, u]$ of the probability distribution $\hat{y}_i^t$. We formulate this process as

$$
u, l = \sigma(f_{\text{sub}}(\hat{y}_i^t \mid sort(\hat{y}_i^t) \mid \hat{y}_i^s)),
\tag{6}
$$

where $\sigma(\cdot)$ is the SIGMOID activation, $sort(\cdot)$ is the decending sort operation, and $\mid$ represents the concatenation operation, $l$ is clipped into the range $[0, u]$.

The predicted quantiles $u$ and $l$ are used to adaptively clip out the high-density semantic classes from the teacher's probability vector $\hat{y}_i^t$. We utilize the clipped high-density classes and the target class with the most probability value to construct a new probability vector $\hat{y}_i^{t*}$, which is formulated as

$$
\hat{y}_i^{t*} = \left[ \left\{ \sigma\left(\frac{v_i^t - l}{\epsilon}\right) \times \sigma\left(\frac{u - v_i^t}{\epsilon}\right) \right\}_{1 \le i \le M} \mid \max(v_1^t, v_2^t, \ldots, v_M^t) \right],
\tag{7}
$$

where $\epsilon = 1e - 7$ is a small positive number that controls smoothness. This implementation ensures proper gradient backpropagation by leveraging the smooth characteristics of the sigmoid function. Specifically, we calculate a mask for the LLM's probability distribution to determine the clipping region (high-density classes), ensuring that gradients can flow correctly. By using this approach, we ensure that the learnable parameters for predicting the lower limit quantile and upper limit quantile are updated correctly during the training process, and gradients are propagated effectively.

The high-density classes and the target class contain the most knowledge in the teacher's probability distribution. Based on the corresponding positions of the clipped classes and target class of $\hat{y}_i^{t*}$, we

Figure 4: The workflow of the span-level correlation distillation. ∘ denotes Hadamard multiplication.

construct the student's new probability vector $\hat{y}_i^{s*}$. Then, we adopt a vanilla KLD to calculate the sum of token-wise distillation loss and the final loss is calculated on the dataset $(X,Y)$:

$$\mathcal{L}_{\text{DAC-KLD}} = E_{x \sim X}\left[\frac{1}{L_{y^{s*}}} \sum_{i=1}^{L_{y^{s*}}} \hat{y}_i^{t*} log \frac{\hat{y}_i^{t*}}{\hat{y}_i^{s*}}\right], \tag{8}$$

where $L_{y^{s*}}$ is the length of the sequence generated from the proposed SCRG strategy.

### 4.3 SPAN-LEVEL CORRELATION CONSISTENCY

Motivated by the work (Liu et al., 2022), we utilize the pre-defined chunker (Kiss & Strunk, 2006) to extract spans (including noun phrases, verb phrases, and prepositional phrases) that have complete meanings from the input sequences, which split a sequence into several token sets. For each token in the input sequence, LLMs predict a high-dimensional probability vector. The relations between tokens within the same span should maintain consistent relations in the transformed probability space. Constraining the relation consistency between the outputs of the student and the teacher models is crucial to transfer semantic knowledge, as shown in Fig. 4.

We divide a probability sequence $[\hat{y}_1, \hat{y}_2, ..., \hat{y}_n]$ into $n_s$ spans $s = [s_1, s_2, ..., s_{n_s}]$ according to the pre-defined span priors from $[y_1, y_2, ..., y_n]$. Here, $s_i = \left[\hat{y}_j, \hat{y}_{j+1}, ..., \hat{y}_{j+n_{s_i}-1}\right]$ represents $i^{th} span$, which starts at the $j^{th}$ token of the sequence and contains $n_{s_i}$ tokens. Both the student and teacher model outputs adhere to the same span priors for token divisions. Consequently, we divide the probability outputs of the student and teacher models into spans, denoting the $i^{th}$ span as

$$s_i^s = \left[\hat{y}_j^s, \hat{y}_{j+1}^s, ..., \hat{y}_{j+n_{s_i}-1}^s\right], s_i^t = \left[\hat{y}_j^t, \hat{y}_{j+1}^t, ..., \hat{y}_{j+n_{s_i}-1}^t\right]. \tag{9}$$

Next, we calculate the correlation between two adjacent tokens within the same spans and ensure consistency of this correlation between the probability outputs of the student model and the teacher model. To achieve this, we utilize the L2 distance to align the consistency. The span consistency loss is defined as follows:

$$\mathcal{L}_{\text{span}} = E_{x \sim X}\left[\frac{1}{n_s} \sum_{i=1}^{n_s} \frac{1}{n_{s_i}} \sum_{(\hat{y}_j^s, \hat{y}_{j+1}^s) \in s_i^s, (\hat{y}_j^t, \hat{y}_{j+1}^t) \in s_i^t} \left\| \hat{y}_j^s \circ \hat{y}_{j+1}^s - \hat{y}_j^t \circ \hat{y}_{j+1}^t \right\|_2\right], \tag{10}$$

where $|\cdot|_2$ represents the L2 distance function, and ∘ denotes the Hadamard multiplication operation calculating correlation in the high-dimensional probability space. It is important to note that the output sequence is also generated by the student using the SCRG strategy. For simplicity, we adopt a standard symbol representation for $\hat{y}_j^t$ and $\hat{y}_j^s$ instead of $\hat{y}_j^{t*}$ and $\hat{y}_j^{s*}$.

### 4.4 OVERALL OPTIMIZATION

We use the proposed KD method in the SFT stage of based models. The student model is supervised by the distillation loss, guided by the finetuned teacher model, and also supervised by the SFT loss. The overall optimization objective for the student model is formulated as

$$\mathcal{L}_{overall} = \mathcal{L}_{\text{SFT}} + \mathcal{L}_{\text{DAC-KLD}} + \mathcal{L}_{\text{span}}. \tag{11}$$

where $\mathcal{L}_{\text{SFT}}$ represents the SFT loss, $\mathcal{L}_{\text{DAC-KLD}}$ represents the distillation loss using the DAC-KLD object, and $\mathcal{L}_{\text{span}}$ represents the span consistency loss which assists the distillation process.

## 5 EXPERIMENTS

In this section, we experiment by initially fine-tuning a large model on the dataset comprising instructions and corresponding responses $(X, Y)$, establishing it as the teacher model $p$. Subsequently, we examine various knowledge distillation methods for distilling a smaller student model under the guidance of the teacher, evaluating the instruction-following performance of the distilled model.

### 5.1 EXPERIMENTAL DESCRIPTION

**Dataset and evaluation metrics.** We conduct the KD experiments on five instruction-following datasets: (1) Dolly Evaluation (Dolly, 2023) is a a sampled subset of atabricks-dolly-15k [1] (Dolly) dataset consists of 500 samples. It covers various behavioural categories such as brainstorming, classification, closed QA, generation, information extraction, open QA, and summarization; (2) Self-Instruct (Wang et al., 2022a) is a dataset for language models' ability to understand and follow instructions. It incorporates 252 expert-written tasks; (3) Vicuna (Wang et al., 2022b) is a dataset consisting of 80 challenging questions used for evaluating the Vicuna model. It follows the evaluation methodology introduced by MiniLLM (Gu et al., 2023); (4) Super-Natural Instruction (Wang et al., 2022b) is introduced as a benchmark, and this dataset contains 1,616 diverse NLP tasks along with their expert-written instructions. It covers 76 different task types, and its test set consists of 9K samples from 119 tasks; (5) Unnatural Instruction (Honovich et al., 2022) dataset comprises 240K instructions generated by AI with minimal human involvement. It shows that AI-generated data can be as effective as human-created data for training language models. The core component of this dataset has 60K samples.

We use the ROUGE-L (Lin, 2004) metric to evaluate the model-generated results and report the average scores of 5 generations for each prompt with different random seeds (10, 20, 30, 40, 50) for all test datasets. ROUGE-L evaluates the precision of the model's output by measuring the longest common subsequence between the generated text and the reference text. It is well-suited for large-scale instruction-following evaluation due to its ability to capture both sentence-level structure and content.

**Base models and baselines.** We distil four kinds of teacher-student model pairs with different model sizes: LLAMA2 (Touvron et al., 2023) (13B teacher, 7B student), OpenLLAMA2 (Geng & Liu, 2023) (7B teacher, 3B student), OPT (Zhang et al., 2022) (6.7B teacher, 1.3B student), GPT2 (Radford et al., 2019) (1.5B teacher, 0.1B student).

We benchmark our method against several advanced knowledge distillation methods: (1) SFT Fine-tunes the student model on a fixed dataset in a vanilla manner; (2) KD (Hinton et al., 2015) utilizes KLD on a fixed dataset; (3) SeqKD (Kim & Rush, 2016) fine-tunes on a teacher-generated dataset; (4) ImitKD (Lin et al., 2020) utilizes KLD on a dataset generated by the student model; (5) GKD (Agarwal et al., 2024) utilizes Jensen-Shannon Divergence (JSD) (Agarwal et al., 2024) on a mixture of a student-generated dataset and a fixed dataset; (6) MiniLLM (Gu et al., 2023) utilizes a policy gradient approach on a dataset generated by the student model; (7) DistiLLM (Ko et al., 2024) utilizes Skew KLD on a student-generated dataset sampling with an off-policy scheduler.

All of our baseline experiments are re-implemented using the open-source code [2] on the same GPU servers utilized by our method. Additionally, we execute these experiments using the exact hyper-parameters as specified in the original codebase.

**Training details.** We follow MiniLLM (Gu et al., 2023) to finetune base models using the training set of the databricks-dolly-15k. Dolly is divided into 14K samples as the training set and equally left 500 samples for validation and testing, respectively. After the fine-tuning process, we select the best-performing model based on its validation set of the Dolly dataset. We then proceeded to test this selected model on the test sets of the five above-mentioned datasets.

For training the teacher and student models, we utilize four A100 (40GB) GPUs for the OpenL-LAMA2, OPT, and GPT2 models and four A800 (80GB) GPUs for the LLAMA2 models. A fixed learning rate of 5e-4 is applied consistently across all experiments. Specifically, for the LLAMA2, OpenLLAMA2, and OPT models, we follow DistiLLM (Ko et al., 2024), employing low-rank adap-

---

[1]https://github.com/databrickslabs/dolly/tree/master
[2]https://github.com/jongwooko/distillm

tation (LoRA) for the query and value weights with a rank of 16 for 10 epochs. In contrast, for the GPT2 models, we fine-tune all parameters for 20 epochs.

Table 1: Comparison of state-of-the-art knowledge distillation methods evaluated by the ROUGE-L metric (Lin, 2004). 'Average' is the average score on the five test datasets The bold and underlined markings signify the best and second-best results, respectively.

| Methods | | Parameters | Datasets | | | | | |
|---|---|---|---|---|---|---|---|---|
| | | | Dolly Evaluation | Self-Instruct | Vicuna | Super-Natural | Unnatural | Average |
| LLAMA2 | Teacher (SFT) | 13B | 29.8241 | 21.0617 | 19.4909 | 35.8318 | 35.7802 | 28.3978 |
| | SFT | 7B | 27.3504 | 28.4430 | 18.7567 | 28.4430 | 30.2788 | 26.6544 |
| | KD (Hinton et al., 2015) | | 27.0737 | 20.7076 | 17.9850 | 30.3350 | 31.4926 | 25.5188 |
| | SeqKD (Kim & Rush, 2016) | | 26.2689 | 19.0278 | 18.4602 | 25.9461 | 28.1010 | 23.5608 |
| | ImitKD (Lin et al., 2020) | | 27.4359 | 20.6792 | 18.8058 | 29.1726 | 30.5764 | 25.3340 |
| | GKD (Agarwal et al., 2024) | | 28.4662 | 22.1717 | 20.7564 | 33.3325 | 33.2682 | 27.5990 |
| | MiniLLM (Gu et al., 2023) | | 30.6447 | 23.9493 | **22.3010** | 34.3454 | 36.0828 | 29.4646 |
| | DistiLLM (Ko et al., 2024) | | 30.7277 | 25.2181 | 20.8356 | 36.1154 | 37.5072 | 30.0808 |
| | Ours | | **31.9195** | **25.4937** | 21.7810 | **37.9154** | **38.1257** | **31.0471** |
| OpenLLAMA2 | Teacher (SFT) | 7B | 27.5100 | 17.9400 | 17.6900 | 32.7500 | 31.4000 | 25.4580 |
| | SFT | 3B | 24.4000 | 16.1300 | 16.5600 | 27.4862 | 28.0500 | 22.5252 |
| | KD (Hinton et al., 2015) | | 25.4814 | 19.1805 | 16.6562 | 31.3307 | 31.8136 | 24.8924 |
| | SeqKD (Kim & Rush, 2016) | | 24.8184 | 16.0980 | 17.2718 | 29.4081 | 28.7395 | 23.2672 |
| | ImitKD (Lin et al., 2020) | | 25.3600 | 18.1600 | 17.5700 | 31.0900 | 28.9600 | 24.2280 |
| | GKD (Agarwal et al., 2024) | | 26.8525 | 20.1060 | 18.4337 | 34.4383 | 32.4797 | 26.4621 |
| | MiniLLM (Gu et al., 2023) | | 28.4950 | **21.7770** | **20.6260** | 35.4001 | 34.7011 | 28.1999 |
| | DistiLLM (Ko et al., 2024) | | 27.8546 | 19.3456 | 19.1723 | 34.4973 | 34.9434 | 27.1627 |
| | Ours | | **29.3062** | 20.5835 | 19.0086 | **37.6171** | **37.2410** | **28.8724** |
| OPT | Teacher (SFT) | 6.7B | 25.8758 | 14.8408 | 16.4199 | 24.9551 | 25.8377 | 21.5859 |
| | SFT | 1.3B | 22.7595 | 11.9784 | 15.2267 | 22.8556 | 24.5763 | 19.4793 |
| | KD (Hinton et al., 2015) | | 22.4476 | 13.4676 | 13.9975 | 23.7679 | 25.4132 | 19.8188 |
| | SeqKD (Kim & Rush, 2016) | | 22.4556 | 12.1588 | 14.8157 | 21.4574 | 24.5907 | 19.0956 |
| | ImitKD (Lin et al., 2020) | | 21.6624 | 12.9286 | 15.8039 | 22.0426 | 24.9619 | 19.4799 |
| | GKD (Agarwal et al., 2024) | | 22.5062 | 12.8309 | 15.3303 | 23.8537 | 26.6441 | 20.2330 |
| | MiniLLM (Gu et al., 2023) | | 24.3168 | 13.5880 | **17.4633** | 26.6789 | 28.7968 | 22.1688 |
| | DistiLLM (Ko et al., 2024) | | 24.7311 | 14.9932 | 16.3293 | 27.1037 | 29.3285 | 22.4972 |
| | Ours | | **27.1486** | **17.3016** | 14.8491 | **32.0618** | **34.9709** | **25.2664** |
| GPT2 | Teacher (SFT) | 1.5B | 27.0357 | 14.5594 | 16.7390 | 24.9659 | 29.4874 | 22.5575 |
| | SFT | 0.1B | 23.8269 | 9.6682 | 14.9022 | 16.4117 | 18.3221 | 16.6262 |
| | KD (Hinton et al., 2015) | | 23.2172 | 10.0899 | 14.9954 | 15.4826 | 18.9597 | 16.5490 |
| | SeqKD (Kim & Rush, 2016) | | 23.7248 | 10.3935 | 14.6558 | 19.8119 | 22.7425 | 18.2657 |
| | ImitKD (Lin et al., 2020) | | 21.7724 | 10.1876 | 15.4640 | 17.1918 | 20.8907 | 17.1013 |
| | GKD (Agarwal et al., 2024) | | 23.3150 | 10.3364 | 15.9384 | 16.0802 | 17.7699 | 16.6880 |
| | MiniLLM (Gu et al., 2023) | | 23.8142 | 12.2771 | 17.0158 | 23.8555 | 24.9101 | 20.3745 |
| | DistiLLM (Ko et al., 2024) | | 25.6114 | 12.5988 | 16.7521 | 24.6374 | **27.5827** | 21.4365 |
| | Ours | | **26.5614** | **13.1174** | **17.6781** | **24.6973** | 27.4025 | **21.8913** |

## 5.2 COMPARISON WITH STATE-OF-THE-ART KD METHODS

We present the quantitative comparison of state-of-the-art knowledge distillation methods evaluated using the ROUGE-L metric in Table 1. It is observed that:

(1) Our method outperforms existing methods in most distillation tasks, with only a few achieving second-best results, across five test datasets, including the LLAMA2, OPT, OpenLLAMA2, and GPT2 series of large language models. Particularly for the OPT datasets, our method shows an average score improvement of over 12% compared to the second-best performing methods.

(2) The KD methods, such as GKD, MiniLLM, and DistiLLM, utilizing student-generated datasets show a greater improvement in enhancing student performance compared to those using the fixed dataset. Furthermore, the distilled student models generally outperform the teacher models, which can be attributed to the mismatch between teacher-forcing training and free-run generation, i.e., exposure

Table 2: Ablation study of the proposed multi-granularity semantic revision.

| Sequence-correcting | DAC-KL | Span Relation | Dolly Validation | Dolly Evaluation | Self-Instruct |
|:---:|:---:|:---:|:---:|:---:|:---:|
| ✗ | ✗ | ✗ | 29.1874 | 24.1603 | 14.8578 |
| ✓ | ✗ | ✗ | 29.6982 | 24.5307 | 14.9485 |
| ✓ | ✓ | ✗ | 30.3486 | 26.9012 | 17.2392 |
| ✓ | ✓ | ✓ | **31.2575** | **27.1486** | **17.3016** |

bias (Bengio et al., 2015). Our method can improve the performance of all student models on average scores of the five test datasets by at least 15%.

(3) We also provide some representative instruction-following cases in Appendix F, further highlighting the effectiveness and superiority of our method in achieving high-quality answers.

## 5.3 ABLATIONS AND ANALYSIS

We provide more ablations and analysis of the proposed methods on the Dolly Validation set, Dolly Evaluation set and Self-Instruct dataset.

**Overall Ablation.** We conduct an overall ablation study to validate the effectiveness of the proposed multi-granularity semantic revision, in Table 2. Initially, employing sequence correction alone yields moderate performance improvement across all evaluation datasets compared to the vanilla result. Upon the addition of DAC-KL, an improvement is observed. A further enhancement is achieved with the inclusion of span-level relation distillation, resulting in more notable performance gains. The most significant improvement is witnessed when all components of the proposed method are combined, leading to the highest performance metrics across all evaluation datasets. This demonstrates that each component contributes to the overall enhancement of model performance, with the combined approach yielding the most substantial improvements. For the span relation loss, we further provide detailed analyses for the span loss and provide examples of the enhanced performance of distilled models relative to their predistillation counterparts in Appendix C.

**Different student-generation methods.** To validate the effectiveness of the proposed SCRG strategy, we compare it with different student-generation methods for sampling the distillation dataset. As illustrated in Table 3a, we observe substantial performance enhancements with SCRG compared to existing student-generation methods. For on-policy sampling, We follow GKD (Agarwal et al., 2024) to utilize a mixture of student-generated and fixed datasets. For off-policy sampling, we follow Distillm (Ko et al., 2024) to adopt an adaptive student-generation schedule for improved sample efficiency. Remarkably, when employing both off-policy and on-policy sampling methods, SCRG achieves notably higher scores across all evaluation metrics. This underscores the effectiveness of SCRG in augmenting performance by improving the quality of generated sequences. Additionally, we provide analysis on the example of SCRG in Appendix E.

**Different distillation loss functions.** To validate the effectiveness of the proposed DAC-KL loss, we compare it with different loss functions in Table 3c. The results demonstrate that DAC-KL significantly outperforms other loss functions across all evaluation metrics. This indicates that DAC-KL effectively captures high-density semantic regions of the teacher's output probability distribution, facilitating easier imitation by the student models. Additionally, we provide the analysis on the visualized examples of the DAC-KL impact on the probability distribution of the teacher's output depicted using kernel density estimation in Appendix D.

**Different components involved in DAC-KL.** The DAC-KL loss guides the distillation process to effectively transfer knowledge from the high-density semantic classes and the target class of the teacher's probability outputs. As evidenced by the results in Table 3b, when both high-density and target classes are considered, the DAC-KL loss achieves the highest validation, evaluation, and self-instruct scores compared to other configurations. This indicates that focusing on these specific classes leads to better performance in knowledge distillation, highlighting the importance of targeting relevant semantic regions for the effective transfer of knowledge.

**Exposure Bias Comparison.** We provide a comparison of our method with existing methods on the exposure bias metric (Gu et al., 2023) in Table 4a. The results show that our method introduces

Table 3: Ablation studies on the proposed SCRG strategy and the DAC-KL loss.

(a) Different student-generation methods

| Generation | Validation | Evaluation | Self-Instruct |
|---|---|---|---|
| On-policy (Lin et al., 2020) | 30.3786 | 26.0948 | 16.1853 |
| Mixed (Agarwal et al., 2024) | 30.8335 | 26.4667 | 16.7789 |
| Off-policy (Ko et al., 2024) | 30.4539 | 27.0961 | 16.7745 |
| SCRG w. off policy | 31.0444 | 27.1453 | 17.2574 |
| SCRG w. on policy | **31.2575** | **27.1486** | **17.3016** |

(b) Components involved in DAC-KL losses

| High-density | Target | Validation | Evaluation | Self-Instruct |
|---|---|---|---|---|
| ✗ | ✓ | 29.3490 | 24.3130 | 14.3810 |
| ✓ | ✗ | 21.3936 | 19.5050 | 11.5035 |
| ✓ | ✓ | **31.2575** | **27.1486** | **17.3016** |

(c) Different distillation loss functions

| Loss Function | Validation | Evaluation | Self-Instruct |
|---|---|---|---|
| Forward KL | 28.9631 | 24.1922 | 14.5108 |
| Reverse KL | 30.0209 | 25.6688 | 14.7184 |
| Symmetric KL | 30.2584 | 27.0961 | 16.7745 |
| Generalized JSD | 27.8759 | 23.3144 | 14.3154 |
| TVD (Wen et al., 2023) | 30.1973 | 25.0033 | 14.6138 |
| SRKL (Ko et al., 2024) | 29.9858 | 25.4849 | 14.9514 |
| SFKL (Ko et al., 2024) | 29.1226 | 25.1400 | 14.4412 |
| DAC-KL | **31.2575** | **27.14864** | **17.3016** |

Table 4: Comparison of different methods on generation length, training efficiency, and regeneration frequency of SCRG.

(a) Exposure bias evaluated by training-decoding discrepancy (ExAccErr) accumulated with generation length. Lower ExAccErr indicates less exposure bias.

| Generation Length | MiniLLM | DistiLLM | Ours |
|---|---|---|---|
| 50 | 6% | 4% | **4%** |
| 100 | 19% | 18% | **16%** |
| 200 | 21% | 20% | **18%** |

(b) Training efficiency for different distillation methods.

| Method | batch(4 samples) / Seconds | Average Rouge-L |
|---|---|---|
| MiniLLM | 0.05 | 28.1999 |
| DistiLLM | 0.25 | 27.1627 |
| Ours w/o SCRG | 0.25 | 28.0122 |
| Ours | 0.18 | **28.6114** |

(c) Impact of SCRG frequency on Average Rouge-L.

| Frequency of SCRG | 0 | 1 | 3 | 5 |
|---|---|---|---|---|
| Average Rouge-L | 28.2016 | 28.8724 | 28.9100 | 28.9710 |

less exposure bias than baselines, by comparing the excess error caused by the training-decoding discrepancy (ExAccErr) accumulated with the generation length. This analysis explains why the distilled student models generally outperform the teacher models.

**Training Efficiency.** Knowledge distillation methods that rely on student-generated output can significantly extend training time, including existing methods MiniLLM and Distillm. However, our approach flexibly combines existing on-policy and off-policy generation sampling methods to balance performance and training efficiency optimally. As evidenced in the Table 4b, our SCRG strategy, when combined with off-policy sampling, achieves superior performance with efficient training for the OpenLLAMA2-3B model on four A800 GPUs.

**Frequency of SCRG.** we provide results from experiments on OpenLLAMA2-3B where multiple SCRGs per sample were conducted, as shown in Table 4c. Considering the trade off between the training cost and performance, we perform sequence correction and re-generation (SCRG) only once per sample, which marginally increases the training time. We also provide more analysis in Appendix E.3

## 6  CONCLUSION

In this paper, we address the challenges in knowledge distillation for LLMs by proposing a novel multi-level semantic revision approach at the sequence, token, and span levels. At the sequence level, our sequence correction and re-generation strategy improves reliability in student-generated sequences. At the token level, the DAC-KL loss function targets semantically salient regions in the teacher's probability distribution, filtering out redundant information. At the span level, input span priors ensure consistent transfer of semantic information across related tokens. Our experiments with four various model series, demonstrate the effectiveness of our approach, significantly improving student model performance over existing KD methods.

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

## A  SOCIAL IMPACT

The primary objective of this study is to contribute to the advancement of the field of Machine Learning, without explicitly emphasizing any specific societal consequences. Although smaller models can lead to positive outcomes, such as reduced emissions, it is crucial to conduct a comprehensive study on potential biases associated with model compression. However, there are potential negative impacts to consider. Model compression may inadvertently exacerbate existing biases within data, leading to unfair outcomes, particularly for underrepresented groups. Additionally, the simplification involved in compression could result in the loss of critical nuances and reduce the model's ability to handle complex tasks accurately.

## B  LIMITATIONS

Our experiments and evaluations were conducted within specific linguistic domains, which may limit the direct applicability of our findings to other domains or tasks. Further research is necessary to determine the generalizability of our approach across various contexts. Additionally, our method's reliance on an external chunker for span extraction could be a limiting factor, especially for low-resource languages where such tools may not be as accessible or effective.

For mainstream languages, however, there is a robust ecosystem of NLP toolkits, such as SpaCy and NLTK, which offer reliable chunking capabilities. These tools have been extensively developed and optimized, ensuring their effectiveness and broad applicability to tasks similar to ours.

For low-resource languages, we suggest that our approach could be adapted by utilizing alternative span extraction methods. For example, in the case of Chinese, the JieBa library effectively enables the extraction of spans, such as noun and verb phrases. For languages with smaller datasets or those that are less resourced, employing large pretrained models like GPT-4 for data preprocessing to generate spans could be a feasible solution. This unsupervised or weakly supervised approach could enhance the adaptability of our method to diverse linguistic resources.

We plan to explore these possibilities in future work, with the aim of broadening the applicability of our method and overcoming its current limitations to extend its utility across different language environments.

## C  DETAILED ANALYSIS OF THE SPAN-LEVEL LOSS FUNCTION.

Our method emphasizes distilling correlation consistency among tokens within a span, rather than merely aligning semantics at the token level (as done in token-level KL divergence). This distinction is critical, as it enables us to capture the nuanced relationships and dependencies inherent within spans. By leveraging the structural and prior knowledge embedded in spans, our method goes beyond simple semantic alignment, enabling more contextually aware and meaningful distillation.

To further clarify the intuition behind our approach, we conducted the following analysis:

### C.1  HUMAN EVALUATION

We compared our Span-Relation method with a random chunking approach (where the number of chunks is controlled to match that of span-relation) and a method that directly extracts relations between adjacent tokens without chunking.

To conduct a more comprehensive and reliable evaluation, we further employed GPT-4 to conduct a human-like evaluation of the models on the Dolly evaluation dataset. We sampled 100 test examples from both models with and without span-level loss and assessed their outputs based on the following criteria:

- **Accuracy (Rate 1-5)**: Does the output correctly include all relevant details from the input?
- **Completeness (Rate 1-5)**: Does the output provide a comprehensive list or description as required by the instruction?
- **Fluency (Rate 1-5)**: Is the output natural, readable, and grammatically correct?
- **Relevance (Rate 1-5)**: How well does the output align with the specific requirements of the instruction?

Table 5: Evaluation Results of Different Loss Types

| Loss Type | Average GPT-4 Evaluation | Dolly Validation | Dolly Evaluation | Self-Instruct |
|---|---|---|---|---|
| w/o Span-Relation loss | 3.89 | 30.3486 | 26.9012 | 17.2392 |
| Adjacent Relation | 4.10 | 30.8348 | 27.0384 | 17.2144 |
| Random Chunking Relation | 4.01 | 30.5938 | 26.9284 | 17.0028 |
| Span-Relation | **4.42** | **31.2575** | **27.1486** | **17.3016** |

These results in Table 5 illustrate that models utilizing span-level loss achieve higher average evaluations across all criteria compared to other configurations, highlighting the benefits of incorporating span-level signals to enhance model performance. Span Priors primarily enhance the semantic coherence of the model's outputs. To evaluate this, we included GPT-4-based human evaluations in our experiments, where the scores were significantly different—*3.89 (w/o Span-Relation Loss) vs. 4.42 (Span-Relation)*—despite the relatively smaller differences observed on datasets like Dolly.

## C.2  ANALYSIS OF THE SPAN-LEVEL LOSS FUNCTION

To demonstrate the improved correlation among span-level tokens, we selected several typical examples where information extraction requires the output to repeat specific phrases from the input prompt. Below, we give an example of the model's performance with and without the span-level loss. Our model distilled with the span-level loss successfully and accurately outputs the fixed phrases from the prompt and context, showcasing the enhanced correlation among tokens within the span.

Table 6: Examples of the enhanced performance of distilled models relative to their predistillation counterparts.

---

**Example 1:**

**Instruction:** Extract the essential tools for a mini survival kit from the following Wikipedia article.

**Input:** Mini survival kits or "Altoids" tin survival kits are small kits that contain a few basic survival tools. These kits often include a small compass, waterproof matches, minimum fishing tackle, large plastic bag, small candle, jigsaw blade, craft knife or scalpel blade, and/or a safety pin/s. Pre-packaged survival kits may also include instructions in survival techniques such as fire-starting or first aid methods. In addition, paracord can be wrapped around the tin. The paracord can be used for setting up an emergency shelter or snaring small animals. They are designed to fit within a container roughly the size of a mint tin.

**Without Span-Level Loss:** The essential tools include a small compass, waterproof matches, fishing tackle, plastic bag, candle, blade, knife, and safety pin.

**Adjacent Relation:** The essential tools include a small compass, waterproof matches, fishing tackle, plastic bag, candle, scalpel blade, and safety pin.

**With Span-Level Loss:** The essential tools include a small compass, waterproof matches, minimum fishing tackle, large plastic bag, small candle, jigsaw blade, craft knife or scalpel blade, and safety pin/s.

**Example 2:**

**Instruction:** Given the reference text below, what is Isthmohyla debilis?

**Input:** Isthmohyla debilis is a rare species of frog in the family Hylidae. It occurs in the Atlantic slopes of the Cordillera Central and Cordillera Talamanca in Costa Rica and western Panama as well as on the Pacific slopes in southwestern Panama. Common name Isla Bonita treefrog has been suggested for it (the type locality is "Isla Bonita" in Costa Rica).

**Without Span-Level Loss:** Isthmohyla debilis is a rare frog species in the family Hylidae. It is found in Costa Rica and Panama. The common name Isla Bonita treefrog is suggested for it.

**Adjacent Relation:** Isthmohyla debilis is a rare species of frog in the family Hylidae. It is found in the Cordillera Central and Costa Rica. The common name Isla Bonita treefrog has been suggested for it.

**With Span-Level Loss:** Isthmohyla debilis is a rare species of frog in the family Hylidae. It occurs in the Atlantic slopes of the Cordillera Central and Cordillera Talamanca in Costa Rica and western Panama as well as on the Pacific slopes in southwestern Panama. The common name Isla Bonita treefrog has been suggested for it.

---

The model distilled with span-level loss can better extract the important adjectives in phrases like minimum fishing tackle, large plastic bag, and small candle in the first example, and output long phrase of complex location as in the second example.

## D ANALYSIS ON THE VISUALIZED PROBABILITY DISTRIBUTION OF THE TEACHER MODEL

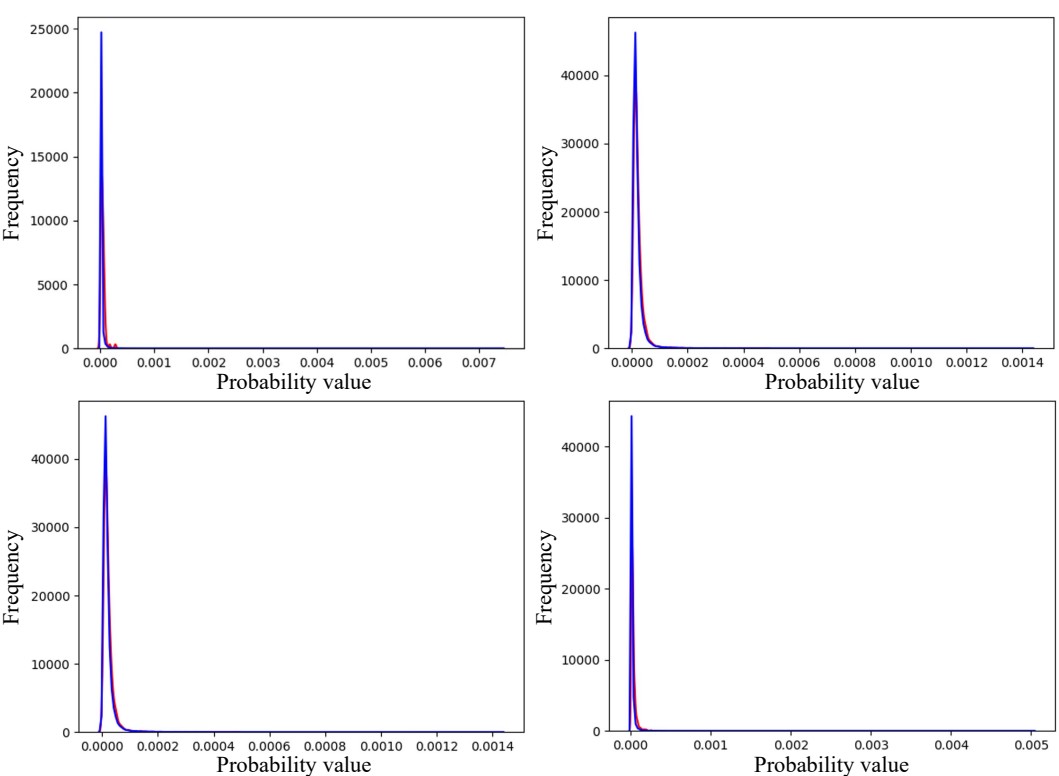

Figure 5: Examples of the probability distribution of the teacher's output are depicted using kernel density estimation. The original distribution is represented by the blue line, while the distribution of the adaptively clipped probability classes is shown by the red line. From this picture, we can observe that the DAC-KL loss constrains the regions of probability distribution with dense semantic knowledge. Enforcing student model to imitate the distribution of these regions can effectively mitigate the training interference caused by low-semantic regions for student models with limited learning capacity.

In Figure 5, We illustrate examples of the teacher's output probability distribution using kernel density estimation. The DAC-KL loss primarily focuses on capturing low-probability yet high-frequency regions of the distribution and combines these with the target class to form new logit vectors.

### D.1 COMPARING OTHER LOGITS-SELECTIVE METHODS WITH DAC LOSS

Our motivation stems from the idea of modulating the probability distribution to reduce the alignment difficulty between the teacher and student distributions, similar to methods like DKD and SKD. However, the key difference is that we suppress redundant information in the original distribution to reduce the difficulty of fitting the student's output to the teacher's distribution when the student's capacity is limited. we compare DAC-KL with other logits-selective methods, including the Fixed Clipping Threshold approach, which is conceptually similar to the method described in Raman et al. (Raman et al., 2023), except that it uses a cumulative sum threshold of 95% as the clipping up.

The necessity of using DAC-KL lies in its adaptive nature for a complex probability distribution. We have tried other clipping and sampling methods, but these approaches rely on manually set thresholds.

Table 7: Comparison of different knowledge distillation methods.

| Method | Dolly Validation | Dolly Evaluation | Self-Instruct |
|---|---|---|---|
| DKD (Zhao et al., 2022) | 29.7182 | 24.3986 | 15.4907 |
| SKD (Yuan et al., 2024) | 29.9332 | 25.2840 | 15.9172 |
| Fixed clipping threshold | 30.7910 | 26.4911 | 16.5682 |
| Zhang et al. (Zhang et al., 2023) | 29.9443 | 25.3442 | 16.0382 |
| Wang et al. (Wang et al., 2021) | 29.8221 | 25.2321 | 15.9138 |
| Raman et al. (Raman et al., 2023) | 30.6910 | 26.3120 | 16.4839 |
| Ours | **31.2575** | **27.1486** | **17.3016** |

Fixed thresholds during training did not perform as well as our current method, which adapts to the probability distribution of different tokens in different samples throughout the training process.

The results in Table 7 allow for direct comparison and demonstrate how DAC-KL, which operates at the token level, provides superior performance by effectively balancing information retention and noise reduction compared to simplistic logit pruning approaches.

## E  ANALYSIS ON THE EXAMPLE OF THE SEQUENCE CORRECTION AND RE-GENERATION (SCRG) STRATEGY.

The role of SCRG is to mitigate the introduction of errors in the data produced by the student model during the initial phase of the knowledge distillation training. It achieves this by employing the teacher model's guidance to refine the generation process, thereby improving the overall quality of the output. We provide an example of the SCRG strategy in Table 8.

Table 8: Example of the student-generated output sequence using the sequence correction and re-generation strategy. The red token represents the detected position of the error token.

---

**Instruction:** What is the difference between men's and women's lacrosse

**Samples from student:** Men's lacrosse has a limited amount of time to play play play as as as as as as as as as as as as as as as as as as as

**Student's Prediction:** Men's lacrosse is a of of of to play and play play as as as as as as as as as as as as as as as as as

**Teacher's Prediction:** Men's lacrosse is a limited number of movesouts play each each. opposed opposed they a they opposed opposed opposed opposed opposed opposed opposed opposed opposed opposed opposed opposed they a they opposed opposed opposed opposed opposed opposed opposed opposed opposed opposed

**Re-sample:** Men's lacrosse has a limited number of players and women's lacrosse has a maximum number of players.

---

### E.1  EMPIRICAL EVIDENCE FOR SCRG

To provide empirical evidence in terms of distinct n-grams, we present a comparison of two example sentences: one generated early in the distillation process without SCRG (Sentence 1) and another generated after applying SCRG (Sentence 2).

**Sentence 1 (Without SCRG)**: "Men's lacrosse has a limited amount of time to play play play as as as as as as as as as as as as as as as as as as"

- **1-grams**: Total: 31, Unique: 12, Distinct-1: 0.387

- **2-grams**: Total: 30, Unique: 13, Distinct-2: 0.433
- **3-grams**: Total: 29, Unique: 14, Distinct-3: 0.483

**Sentence 2 (With SCRG)**: "Men's lacrosse has a limited number of players and women's lacrosse has a maximum number of players."

- **1-grams**: Total: 19, Unique: 12, Distinct-1: 0.632
- **2-grams**: Total: 18, Unique: 13, Distinct-2: 0.722
- **3-grams**: Total: 17, Unique: 14, Distinct-3: 0.824

The distinct n-gram statistics reveal significant improvements in output quality when SCRG is applied. Sentence 2 exhibits higher distinct n-gram scores across all levels, demonstrating an increase in unique words and phrases. This not only highlights the effectiveness of SCRG in refining data but also emphasizes its role in enhancing the overall quality of the student model's generation process.

### E.2 COMPARISION TO THE LEADING DATA QUALITY IMPROVEMENT APPROACH

Furthermore, we conducted experiments to provide a robust comparison of SCRG against a leading data quality improvement approach by Kim et al. (Kim & Baek, 2024), which focuses on offline data pruning and selection.

Table 9: Performance Comparison of SCRG and Kim et al.'s Method

| Data Enhancement | Dolly Validation | Dolly Evaluation | Self-Instruct |
|---|---|---|---|
| Kim et al. | 30.7346 | 26.8665 | 17.2208 |
| SCRG | 31.2575 | 27.1486 | 17.3016 |
| SCRG + Kim et al. | 31.3610 | 27.2068 | 17.3342 |

These results in Table 9 show that SCRG not only outperforms the approach by Kim et al. but when combined with Kim et al.'s method, a slight improvement in performance is observed. While both SCRG and the method proposed by Kim et al. enhance data quality, the incremental gains from combining them are limited. This is likely due to the fact that both methods address similar underlying issues related to data quality, resulting in diminishing returns when applied together.

### E.3 IMPACT OF THE FREQUENCY OF SCRG

It is important to clarify that the purpose of our corrections is not to make the student model's outputs identical to the teacher's but to provide initial guidance that prevents severe errors, such as repetitive generation. In our experiments, even a single correction significantly improves output quality. While increasing the number of corrections may lead to outputs that resemble those of the teacher, this approach does not effectively address exposure bias. We have conducted a performance experiment on OpenLLAMA2-3B, which demonstrates how varying the number of corrections impacts the results, as shown below:

Table 10: Impact of varying the number of corrections on output quality measured by Average Rouge-L.

| Frequency of SCRG | 0 | 1 | 3 | 5 | 10 |
|---|---|---|---|---|---|
| Average Rouge-L | 28.2016 | 28.8724 | 28.9100 | 28.9710 | 28.3273 |

The results in Table 10 demonstrate the scenario where we directly address exposure bias using student-generated data, which, while effective, can introduce additional generation errors. Our aim is not to ignore the exposure bias but to highlight the challenges inherent in this approach.

To address these challenges and to mitigate the introduction of generation errors, we have implemented SCRG. SCRG is not only about solving exposure bias but also about doing so in a way that avoids

the propagation of erroneous data. It achieves this by refining the student model's outputs with the guidance of the teacher model, thus enhancing the quality of the distillation dataset without compromising the integrity of the data.

Furthermore, the 'Frequency of SCRG=10' experiment was included to illustrate the scenario where student-generated data closely resembles that of the teacher, which, as you correctly pointed out, could potentially undermine the effectiveness of addressing exposure bias. This experiment serves to demonstrate the balance that SCRG strikes between maintaining the teacher's guidance and the student's independence.

## F EXAMPLES OF MODEL RESPONSES ON THE INSTRUCTION-FOLLOWING TASK.

Table 11: Instruction-following cases from the databricks-dolly-15k dataset.

| Case #1, Category: classification | |
|---|---|
| Instruction | If we were playing a game where we had to identify things that can be found inside a house, which of these would we call out: car, chair, table, park, cloud, microwave. |
| Input | - |
| Groundtruth | Inside a house you can find a chair, a table and a microwave. Cars, clouds and parks are found outside of houses. |
| SFT | car, chair, table, microwave. |
| KD | Car, chair, table and microwave are things that can be found in a house. Park and cloud are unlikely to be found in a house but are things that can be seen from a house. |
| SeqKD | We would call out: chair, table, car and microwave. |
| ImitKD | Car, chair, table and microwave are items that can be found inside of a house. Park and cloud are outdoors items. |
| MiniLLM | Car, chair, table, and microwave can all be found inside a house. Park and cloud cannot be found inside a house because they are neither objects nor locations. |
| GKD | You can find a car, chair, table and microwave inside a house. A park and cloud are not found inside a house. |
| DISTILLM | In this list, we can find a car, chair, table and microwave. A park and a cloud are not things you would typically find inside a house. |
| Ours | In this game, a car, park, and cloud would not be things that could be found inside a house. A chair, table, and microwave would be things that could be find inside a house. |

Table 12: Instruction-following cases from the databricks-dolly-15k dataset.

| Case #2, Category: closed QA | |
|---|---|
| Instruction | How many managers have Arsenal had? |
| Input | There have been twenty permanent and eight caretaker managers of Arsenal since 1897; Stewart Houston has managed the club in two separate spells as caretaker. The most successful person to manage Arsenal is Arsé8ne Wenger, who won three Premier League titles, seven FA Cups and seven Community Shields between 1996 and 2018. Wenger is the club's longest-serving manager; he surpassed George Allison's record of 13 years in October 2009. Two Arsenal managers have died in the job Ž2013 Herbert Chapman and Tom Whittaker. |
| Groundtruth | Arsenal have had twenty permanent and eight caretake managers. |
| SFT | There have been thirty-seven managers since 1896. |
| KD | There have been 28 managers of Arsenalf FC. |
| SeqKD | Arsenal has had 20 permanent managers, 8 caretaker managers and 2 manager with dead in the job. |
| ImitKD | There have been thirty-seven managers since 1896. |
| MiniLLM | Since 1897, Arsenal have had twenty-eight managers. Eight of these were caretaker managers. The most successful manager is Arséne Wener, who won three Premier Leagues, seven FA Cups, and seven Community Shield trophies. He is the longest-serving manager, as he surpassed George Alison's record of twelve years in October 2,009. Two managers have died in this job; Herbert Chapman and Tom Whitaker. |
| GKD | There have been thirty managers of Arsenals football club since 1896. This includes twenty permanent managers and eight caretaker or interim managers. |
| DISTILLM | There have been 20 permanent managers and 8 caretaker managers for a total of 28 managers. |
| Ours | There have been 20 permanent and 8 caretaker managers at Arsenal. |

Table 13: Instruction-following cases from the databricks-dolly-15k dataset.

| Case #3, Category: brainstorming | |
| --- | --- |
| Instruction | What is a good way to get around without a car? |
| Input | - |
| Groundtruth | If you live in a city you can access public transport such as buses and trains. You can also walk, use a bicycle, or electric scooter to get around. |
| SFT | When you don't have a car, you need to find alternative ways to get around. If you are able-bodied and live in a city, your best bet would be to use public transportation. Every city has trains, buses and other modes of transportation that are accessible without a car.If you are unable-bodied or have a car that doesn't work, you may want to consider ride-sharing services such as Uber and Lyft. |
| KD | A few examples of ways to get around without a personal car are using public transportation such as the bus or subway, carpooling with friends or coworkers, or using ride-sharing services such as Uber or Lyft. If none of these options are viable for you, it is still possible to get around without a vehicle; for example, it is often possible to walk or ride a bike for short distances, and in some cases, it is possible to use a bicycle, scooter, or skateboard for longer distances. |
| SeqKD | There are multiple options for getting around without a car. Biking, walking, riding a scooter, riding a bike with a scooter, using public transportation, rideshare, or a moped are all good options. If you are in a smaller town where public transportation is available, walking or riding a bike may also be a good option. |
| ImitKD | Public transportation is a good way to avoid a need for a car. In many cities in the US, public transportation includes buses, subways, and trains that can take you where you need to go. For shorter distances, many cities and towns have bike-shares which can provide an easy way to get around without having to rely on a car. |
| MiniLLM | There are many ways to get around without a driver's license or car. The most common way is to walk, but you can also use a bicycle, public transportation, or ride-sharing services. If you live in a city, you can also take advantage of public transportation, which is often more affordable than owning a car. |
| GKD | There are multiple ways to get around without a personal vehicle. Public transportation is available in many cities and towns. Many cities also have bike share programs, where you can rent a bike or low-cost electric scooter for short trips. You can also walk, but this may depend on the distance and weather. |
| DISTILLM | There are many ways to get around without cars. You can walk, ride a bike, use public transportation, or ride share services such as Uber and Lyft. |
| Ours | If you don't have a car, there are still many ways to get around. You can take public transportation, which includes buses, trains, and subways. If you are near a large city, you may also have access to light rail, which is like a bus that runs on rails. If you are in a suburb, you can use public transportation such as a bus or train. If you live in a small town, you may need to take a taxi or ride share service. |

# G  PROMPT TEMPLATE FOR THE INSTRUCTION-FOLLOWING TASK

We provide the prompt template for the instruction-following task in Table 14.

Table 14: The prompt template for training and evaluation of instruction-following task experiments.

Below is an instruction that describes a task.
Write a response that appropriately completes the request.

### Instruction:
{instruction}

### Input:
{input}

### Response:

# H    STATISTICAL SIGNIFICANCE TESTS

We appreciate the reviewer's suggestion regarding statistical significance testing. To clarify, our experiments were conducted using 5 random seeds, with the reported results representing the average performance across these runs. While we did not perform formal statistical significance tests, we computed the standard deviations for each result, and based on our observations, there were no large anomalies or outliers in the data. Below, we provide the average values along with the corresponding standard deviations for each metric:

Table 15: Performance comparison with standard deviations.

| Sequence-correcting | DAC-KL | Span Relation | Dolly Validation (↑) | Dolly Evaluation (↑) | Self-Instruct (↑) |
|:---:|:---:|:---:|:---:|:---:|:---:|
| × | × | × | 29.1874 (0.18) | 24.1603 (0.22) | 14.8578 (0.15) |
| ✓ | × | × | 29.6982 (0.19) | 24.5307 (0.21) | 14.9485 (0.16) |
| ✓ | ✓ | × | 30.3486 (0.21) | 26.9012 (0.23) | 17.2392 (0.18) |
| ✓ | ✓ | ✓ | **31.2575** (0.19) | **27.1486** (0.22) | **17.3016** (0.17) |

# I    MORE COMPLEX TEACHER THAN STUDENT

We extended our experiments from the previous OPT 6.7B → 1.3B distillation setup by using a larger teacher model, OPT-13B, to distil the 1.3B student. The results, shown in the table below, demonstrate that while distilling with a much larger teacher does lead to smaller improvements in performance compared to the 6.7B → 1.3B case, our proposed distillation method still outperforms vanilla KD loss significantly, even when using a much more complex teacher.

Table 16: Performance comparison of distillation methods using different teacher models.

| Model | Method | Params | Dolly Evaluation | Self-Instruct | Vicuna | Super-Natural | Unnatural | Average |
|:---|:---|:---:|:---:|:---:|:---:|:---:|:---:|:---:|
| **OPT(6.7B-1.3B)** | Teacher (SFT) | 6.7B | 25.8758 | 14.8408 | 16.4199 | 24.9551 | 25.8377 | 21.5859 |
| | Student (SFT) | 1.3B | 22.7595 | 11.9784 | 15.2267 | 22.8556 | 24.5763 | 19.4793 |
| | Vanilla KD | 1.3B | 22.4476 | 13.4676 | 13.9975 | 23.7679 | 25.4132 | 19.8188 |
| | **Ours** | 1.3B | **27.1486** | **17.3016** | **14.8491** | **32.0618** | **34.9709** | **25.2664** |
| **OPT(13B-1.3B)** | Teacher (SFT) | 13B | 26.4438 | 15.9537 | 17.1171 | 28.1131 | 29.0092 | 23.3274 |
| | Student (SFT) | 1.3B | 22.7595 | 11.9784 | 15.2267 | 22.8556 | 24.5763 | 19.4793 |
| | Vanilla KD | 1.3B | 22.7027 | 12.8890 | 14.8943 | 21.9863 | 25.0162 | 19.4977 |
| | **Ours** | 1.3B | **26.5122** | **15.7949** | **15.6140** | **31.4153** | **34.4243** | **24.7522** |

As seen from Table 16, while the performance improvement decreases with the larger teacher (OPT-13B), our distillation method still provides a significant advantage over the vanilla KD approach, even when using a more complex and larger teacher model. This indicates that our method with DAC-KL loss helps mitigate the potential performance degradation seen when distilling with a much larger teacher.

