# OpenReview forum: "Multi-Granularity Semantic Revision for Large Language Model Distillation"
_ICLR.cc/2025/Conference — Submitted to ICLR 2025_

### Official Review · Reviewer_hJE8 · 2024-11-03

**Soundness:** 3
**Presentation:** 3
**Contribution:** 2
**Rating:** 5
**Confidence:** 3

**Summary:**

The authors propose three separate ideas for knowledge distillation: 1) using student-generated samples for distillation, but also correct them if it’s not the same as the teachers generated, 2) adaptively clipping the distribution before applying the KL loss, and 3) applying a span-level loss, where the goal is to match the between-token correlation within each span.

Update: I followed the authors' arguments for exposure bias, and I would like to thank Reviewer S2Y9 for chiming in and pointing out some flaws in the arguments.

I want to add that I am aware of the scheduled sampling paper. My point was that despite addressing exposure bias, it's not been a popular technique for LLMs, making me believe exposure bias isn't as important as the authors claim. In fact, more recent papers (e.g., [1]) have shown that maybe exposure bias is not a big problem.

Looking at the generations the author showed "Men’s lacrosse has a limited amount of time to play play play as as as as as as as as as as as as as as as as as as as", I feel a big reason for the improvement is that the student is too poor, as it cannot even avoid simple repetitions that should have never occurred in the training data.

Overall, I am raising my score to 5 (i.e., still a bit negative). I think a more detailed analysis of exposure bias can significantly strengthen the paper (e.g., including scheduled sampling as a baseline), as it appears central to the authors' claims.

[1] https://aclanthology.org/2021.emnlp-main.415/

**Strengths:**

1. The authors perform experiments across a wide range of models and datasets.
2. The authors compare with a variety of baselines.

**Weaknesses:**

1. The intuition for correcting the distillation dataset is unclear. If you keep correcting it to be the same as the teacher’s generation, it is almost equivalent to simply using the teacher’s outputs as the distillation dataset. It would be ideal to have a comparison with the teacher's samples (or maybe I missed it).
2. The authors lack detailed analyses about the clipping method. For example, it would be much better if the authors can show what the predicted clipping thresholds are, and how that compares with simply using the mean of these clipping thresholds.
3. No detailed analyses for the span loss. Although the authors show span-level correlation can improve performance in the ablation study, the authors do not study the different designs, e.g., correlation measure, chunking methods, etc.

Overall, I feel the authors propose a wide range of useful (but also somewhat unrelated) techniques to improve KD performance. However, these techniques, individually, are perhaps under-studied.

**Questions:**

The ablation study does not provide a full picture of how important each techniques is. I am curious about how these methods work in separation.

---

> ### Author Response · Authors · 2024-11-18
>
> Below is our point-by-point response to your main concerns. Please let us know if there's anything we can clarify further.
>
> ### **Weakness1: The intuition for correcting the distillation dataset**
> It is important to clarify that the purpose of our corrections is not to make the student model's outputs identical to the teacher's but to provide initial guidance that prevents severe errors, such as repetitive generation. In our experiments, even a single correction significantly improves output quality. While increasing the number of corrections may lead to outputs that resemble those of the teacher, this approach does not effectively address exposure bias. We have conducted a performance experiment on OpenLLAMA2-3B, which demonstrates how varying the number of corrections impacts the results, as shown below:
> | Frequency of SCRG | 0       | 1       | 3       | 5       |10
> |-----------------------------------|---------|---------|---------|---------|---------|
> | Average Rouge-L  | 28.2016 | 28.8724 | 28.9100 | 28.9710 |28.3273
>
>
> ### **Weakness2: Detailed analyses about the clipping method**
> In response to your concern, we would like to clarify that detailed analyses of the clipping method have been provided in **Appendix D**. Specifically, we illustrate examples of the teacher's output probability distribution using kernel density estimation. The DAC-KL loss primarily focuses on capturing low-probability yet high-frequency regions of the distribution and combines these with the target class to form new logit vectors.
>
> Additionally, further discussions are presented in **Appendix I**. Our approach is motivated by the goal of modulating the probability distribution to reduce the alignment difficulty between teacher and student distributions. This is conceptually similar to methods like **DKD** [1] and **SKD** [2]. However, the key distinction lies in how **DAC-KL** adaptively suppresses redundant information in the original distribution by a learnable sub-network. This suppression reduces the challenge of fitting the teacher's distribution when the student's capacity is limited.
>
> The necessity of **DAC-KL** lies in its adaptability to complex probability distributions. While other clipping and sampling methods rely on manually defined thresholds, **DAC-KL** adjusts dynamically to the probability distribution of different tokens across samples during training. Fixed thresholds, though simpler, performed less effectively, as shown in the results below:
>
> | Method                   | Dolly Validation | Dolly Evaluation | Self-Instruct |
> |--------------------------|------------------|------------------|---------------|
> | DKD [1]                 | 29.7182          | 24.3986          | 15.4907       |
> | SKD [2]                 | 29.9332          | 25.2840          | 15.9172       |
> | Fixed clipping threshold | 30.7910          | 26.4911          | 16.5682       |
> | **Ours**                 | **31.2575**      | **27.1486**      | **17.3016**   |
>
> As demonstrated in the table, our approach significantly outperforms fixed clipping thresholds and other baseline methods across all metrics. **DAC-KL**'s adaptive nature enables it to optimize the probability distribution modulation dynamically, which is crucial for effective distillation under limited student capacity.
>
> We appreciate the reviewer’s suggestion and have expanded our discussions to clarify this aspect further. Thank you for highlighting this point.
>
> [1]Zhao B, Cui Q, Song R, et al. Decoupled knowledge distillation[C]//Proceedings of the IEEE/CVF Conference on computer vision and pattern recognition. 2022: 11953-11962.
> [2] Yuan M, Lang B, Quan F. Student-friendly knowledge distillation[J]. Knowledge-Based Systems, 2024, 296: 111915.

---

> ### Author Response · Authors · 2024-11-18
>
> ### **Weakness3: Dailed analyses for the span loss**
>
> ##### **Human Evaluation**
>
> We compared our Span-Relation method with a random chunking approach (where the number of chunks is controlled to match that of span-relation) and a method that directly extracts relations between adjacent tokens without chunking.
>
>
> To conduct a more comprehensive and reliable evaluation, we further employed GPT-4 to conduct a human-like evaluation of the models on the Dolly evaluation dataset. We sampled 100 test examples from both models—with and without span-level loss—and assessed their outputs based on the following criteria:
>
> - **Accuracy (Rate 1-5)**: Does the output correctly include all relevant details from the input?
> - **Completeness (Rate 1-5)**: Does the output provide a comprehensive list or description as required by the instruction?
> - **Fluency (Rate 1-5)**: Is the output natural, readable, and grammatically correct?
> - **Relevance (Rate 1-5)**: How well does the output align with the specific requirements of the instruction?
>
> The evaluation results are summarized in the Table below:
>
> | Loss Type                                 | Average GPT-4 Evaluation |Dolly Validation | Dolly Evaluation | Self-Instruct |
> |-------------------------------------------|--------------------------|------------------|------------------|---------------|
> | w/o Span-Relation loss                      | 3.89                     | 30.3486          | 26.9012          | 17.2392       |
> | Adjacent Relation (w/o Span Priors)      | 4.10                  |30.8348 | 27.0384 | 17.2144 |
> | Random Chunking Relation (w/o Span Priors) | 4.01 |30.5938 | 26.9284 | 17.0028|
> | Span-Relation                             | 4.42                     |31.2575 | 27.1486 | 17.3016 |
>
> These results illustrate that models utilizing span-level loss achieve higher average evaluations across all criteria compared to other configurations, highlighting the benefits of incorporating span-level signals to enhance model performance.
>
> ##### **Example Outputs**
>
> To demonstrate the improved correlation among span-level tokens, we selected several typical examples where information extraction requires the output to repeat specific phrases from the input prompt. Below, we give examples of different models with and without span-level loss. Our model distilled with the span-level loss successfully and accurately outputs the fixed phrases from the prompt and context, showcasing the enhanced correlation among tokens within the span.
>
> **Example 1**
> - **Instruction**: Extract the essential tools for a mini survival kit from the following Wikipedia article.
> - **Input**: Mini survival kits or "Altoids" tin survival kits are small kits that contain a few basic survival tools. These kits often include a small compass, waterproof matches, minimum fishing tackle, large plastic bag, small candle, jigsaw blade, craft knife or scalpel blade, and/or a safety pin/s. Pre-packaged survival kits may also include instructions in survival techniques such as fire-starting or first aid methods. In addition, paracord can be wrapped around the tin. The paracord can be used for setting up an emergency shelter or snaring small animals. They are designed to fit within a container roughly the size of a mint tin.
> - **Without Span-Level Loss**: The essential tools include a small compass, waterproof matches, fishing tackle, plastic bag, candle, blade, knife, and safety pin.
> - **With Span-Level Loss**: The essential tools include a small compass, waterproof matches, minimum fishing tackle, large plastic bag, small candle, jigsaw blade, craft knife or scalpel blade, and safety pin/s.
>
> **Example 2**
> - **Instruction**: Given the reference text below, what is Isthmohyla debilis?
> - **Input**: Isthmohyla debilis is a rare species of frog in the family Hylidae. It occurs in the Atlantic slopes of the Cordillera Central and Cordillera Talamanca in Costa Rica and western Panama as well as on the Pacific slopes in southwestern Panama. Common name Isla Bonita treefrog has been suggested for it (the type locality is "Isla Bonita" in Costa Rica).
> - **Without Span-Level Loss**: Isthmohyla debilis is a rare frog species in the family Hylidae. It is found in Costa Rica and Panama. The common name Isla Bonita treefrog is suggested for it.
> - **With Span-Level Loss**: Isthmohyla debilis is a rare species of frog in the family Hylidae. It occurs in the Atlantic slopes of the Cordillera Central and Cordillera Talamanca in Costa Rica and western Panama as well as on the Pacific slopes in southwestern Panama. The common name Isla Bonita treefrog has been suggested for it.
>
> The model distilled with span-level loss can better extract important adjectives in phrases like "minimum fishing tackle," "large plastic bag," and "small candle" in the first example, and output long phrases of complex location as in the second example.

---

> > ### Author Response · Authors · 2024-11-22
> >
> > Thank you sincerely for your review. We would greatly appreciate it if you could inform us of any remaining questions or concerns that you may have so that we can address them promptly prior to the deadline. Alternatively, if you feel that your initial concerns are addressed, we would appreciate updating your evaluation to reflect that.
> >
> > Thank you!

---

> > ### Author Response · Authors · 2024-11-25
> >
> > If you feel that our responses have sufficiently addressed your initial concerns and that there are no further issues to discuss, we would be immensely grateful for your confirmation. Your prompt response will greatly assist us in moving forward with our work.
> >
> > Thank you very much for your time and consideration.

---

> > > ### Comment · Reviewer_hJE8 · 2024-11-26
> > >
> > > Thank you for the detailed response!
> > >
> > > I still feel my first concern is not fully addressed, partially because the authors presented an alternative experiment than what I suggested.
> > >
> > > Although the authors argue exposure bias is a significant problem, studies also show LLMs generally perform well despite using teacher forcing.
> > >
> > > Overall, I am willing to raise the soundness score for the additional results but maintain my overall assessment.

---

> > > > ### Author Response · Authors · 2024-11-30
> > > > **Additional concerns about exposure bias**
> > > >
> > > > I would like to express my concern regarding your statement: "Although the authors argue exposure bias is a significant problem, studies also show LLMs generally perform well despite using teacher forcing."
> > > >
> > > > In our research, we have extensively reviewed literature that highlights the significant impact of exposure bias on performance. Works like [1] clearly indicate that exposure bias can adversely affect model outputs, which aligns with our findings. Both DistillM and MiniLLM, as well as our own study, consistently recognize exposure bias as a critical issue that must be addressed.
> > > >
> > > > Furthermore, as shown in Table 1 of our main text, methods that do not account for exposure bias—such as SeqKD and ImIKD, which rely solely on teacher forcing—demonstrate markedly poor performance. This empirical evidence reinforces our position that exposure bias is a crucial factor that cannot be overlooked.
> > > >
> > > > [1] Bengio, S., Vinyals, O., Jaitly, N., et al. "Scheduled sampling for sequence prediction with recurrent neural networks." Advances in Neural Information Processing Systems, 2015, 28.
> > > >
> > > > Thank you for considering my request for clarification on this important topic. I look forward to your response!

---

> > > > > ### Comment · Reviewer_S2Y9 · 2024-12-01
> > > > > **Comment on author's response**
> > > > >
> > > > > With all due respect, I must point out that the author’s claim below is flawed:
> > > > >
> > > > > >..methods that do not account for exposure bias—such as SeqKD and ImIKD, which rely solely on teacher forcing
> > > > >
> > > > > In fact, ImitKD exactly addresses the issue of exposure bias and doesn't solely rely on teacher forcing. This is clear even from reading the abstract of the original paper [1].
> > > > >
> > > > > And I don't see how the proposed method eliminates exposure bias. In contrast, it should enlarge (compared to other methods like DistilLLM) the gap between the prefix distributions in training and testing by having the teacher model "correct" the token in the prefix.
> > > > >
> > > > >
> > > > >
> > > > > [1] Lin, Alexander, et al. "Autoregressive Knowledge Distillation through Imitation Learning." Proceedings of the 2020 Conference on Empirical Methods in Natural Language Processing (EMNLP). 2020.

---

> > > > > > ### Author Response · Authors · 2024-12-01
> > > > > >
> > > > > > Thank you very much for your response and your insightful comments!
> > > > > >
> > > > > > Indeed, ImIKD also addresses the important issue of exposure bias. However, we would like to clarify that our approach to tackling exposure bias follows the methodologies of DistillM and MiniLLM by utilizing the student model to generate data for distillation. Unlike these two methods, we have specifically considered the problem of generation errors, as illustrated in Figure 1 of our main text.
> > > > > >
> > > > > > Therefore, we provide an analysis of the impact of correction frequency during the student-generated data process and compare these results with those obtained from using data generated solely by the teacher model.

---

> > > > > > > ### Comment · Reviewer_S2Y9 · 2024-12-02
> > > > > > >
> > > > > > > I agree with your latest response. However, it seems quite different from your earlier reply to Reviewer hJE8, which implied that your method eliminates exposure bias. Instead, your approach addresses the **side effects** of existing methods aimed at eliminating exposure bias, that is, the noise (error tokens) in the student-generated prefix makes the supervision signal from the teacher model unreliable.

---

> > > > > > > > ### Author Response · Authors · 2024-12-03
> > > > > > > >
> > > > > > > > We are extremely grateful for your recognition of our final response.
> > > > > > > >
> > > > > > > > Our SCRG approach, as meticulously detailed in Figure 1 of our main paper, focuses on resolving the additional error generation problems that emerge when existing methods attempt to deal with exposure bias. We do not claim to completely eliminate exposure bias but rather mitigate the side effects and errors that can arise during that process.
> > > > > > > >
> > > > > > > > Regarding the concern raised by hJE8 initially, specifically "It would be ideal to have a comparison with the teacher's samples (or maybe I missed it)", we have addressed this in the experiment provided during our discussion. We presented the scenario where Frequency of SCRG = $\infty$, which indicates the exclusive use of teacher-generated data for distillation. This allows for a direct comparison and offers a more in-depth understanding of how our method fares against the sole use of teacher‘s samples.
> > > > > > > >
> > > > > > > > We sincerely hope that hJE8 will take this clarification into account and re-evaluate our work.

---

> > > > ### Author Response · Authors · 2024-12-01
> > > > **An experiment that utilizes solely the teacher model's samples for distillation**
> > > >
> > > > Thank you for your patience and understanding throughout this review process.
> > > >
> > > > We have specifically included an experiment that utilizes solely the teacher model's samples for distillation, completely excluding any sampling from the student model. This approach provides a clearer comparison and underscores the impact of exposure bias on performance.
> > > >
> > > > | Frequency of SCRG | 0      | 1      | 3      | 5      | 10     | $\infty$     |
> > > > |-------------------|--------|--------|--------|--------|--------|---------|
> > > > | Average Rouge-L   | 28.2016| 28.8724| 28.9100| 28.9710| 28.3273| 26.9084 |
> > > >
> > > > *Note:* $\infty$ represents only using the teacher's samples.
> > > >
> > > >
> > > > When 'Frequency of SCRG=$\infty$', it implies the exclusive use of the teacher's samples. Due to exposure bias, there is a significant degradation in performance. This highlights the importance of striking a balance between the teacher's influence and the student's independent learning to mitigate the adverse effects of exposure bias.

---

> ### Author Response · Authors · 2024-11-27
>
> Thank you for your continued engagement and for considering the additional results we provided. We understand your initial concern and appreciate your willingness to raise the soundness score.
>
> We understand your concern regarding the 'Frequency of SCRG=0' experiment and the potential misunderstanding it may have caused. We wish to emphasize that this experiment was designed to demonstrate the scenario where we directly address exposure bias using student-generated data, which, while effective, can introduce additional generation errors. Our aim is not to ignore the exposure bias but to highlight the challenges inherent in this approach.
>
> To address these challenges and to mitigate the introduction of generation errors, we have implemented SCRG. SCRG is not only about solving exposure bias but also about doing so in a way that avoids the propagation of erroneous data. It achieves this by refining the student model's outputs with the guidance of the teacher model, thus enhancing the quality of the distillation dataset without compromising the integrity of the data.
>
> Furthermore, the 'Frequency of SCRG=10' experiment was included to illustrate the scenario where student-generated data closely resembles that of the teacher, which, as you correctly pointed out, could potentially undermine the effectiveness of addressing exposure bias. This experiment serves to demonstrate the balance that SCRG strikes between maintaining the teacher's guidance and the student's independence.
>
> To provide a more intuitive understanding of SCRG and its impact on the distillation dataset, we have conducted additional analyses, which we detail below:
>
>
> ### **More analyses for correcting the distillation dataset (SCRG)**
> The role of SCRG is to mitigate the introduction of errors in the data produced by the student model during the initial phase of the knowledge distillation training. It achieves this by employing the teacher model's guidance to refine the generation process, thereby improving the overall quality of the output.
>
> To provide empirical evidence, we present a comparison of two example sentences: one generated early in the distillation process without SCRG (Sentence 1) and another generated after applying SCRG (Sentence 2).
>
> ##### Sentence 1 (Without SCRG):
> *"Men’s lacrosse has a limited amount of time to play play play as as as as as as as as as as as as as as as as as as as"*
>
> - **1-grams**:
>   - Total: 31
>   - Unique: 12
>   - Distinct-1: 0.387
> - **2-grams**:
>   - Total: 30
>   - Unique: 13
>   - Distinct-2: 0.433
> - **3-grams**:
>   - Total: 29
>   - Unique: 14
>   - Distinct-3: 0.483
>
> ##### Sentence 2 (With SCRG):
> *"Men’s lacrosse has a limited number of players and women’s lacrosse has a maximum number of players."*
>
> - **1-grams**:
>   - Total: 19
>   - Unique: 12
>   - Distinct-1: 0.632
> - **2-grams**:
>   - Total: 18
>   - Unique: 13
>   - Distinct-2: 0.722
> - **3-grams**:
>   - Total: 17
>   - Unique: 14
>   - Distinct-3: 0.824
>
> The distinct n-gram statistics reveal a significant improvement in generation diversity when SCRG is applied. Sentence 2 demonstrates higher distinct n-gram scores across all levels compared to Sentence 1. This increase in unique words and phrases highlights SCRG’s effectiveness in promoting diverse and meaningful outputs in the student model.
>
> Furthermore, we conducted experiments to provide a robust comparison of SCRG against a leading data quality improvement approach by Kim et al. [1], which focuses on offline data pruning and selection.
>
> [1] Kim M, Baek S. Measuring Sample Importance in Data Pruning for Training LLMs from a Data Compression Perspective[J]. arXiv preprint arXiv:2406.14124, 2024.
>
> #### **Experimental Results**
>
> Our results, summarized in the Table below, demonstrate that SCRG outperforms the offline data enhancement method proposed by Kim et al. across multiple datasets:
>
> | Data Enhancement  | Dolly Validation | Dolly Evaluation | Self-Instruct |
> |-------------------|------------------|------------------|---------------|
> | Kim et al.        | 30.7346          | 26.8665          | 17.2208       |
> | SCRG              | 31.2575          | 27.1486          | 17.3016       |
> | SCRG + Kim et al. | 31.3610          | 27.2068          | 17.3342       |
>
> These results show that SCRG not only outperforms the approach by Kim et al., but when combined with Kim et al.'s method, a slight improvement in performance is observed. While both SCRG and the method proposed by Kim et al. enhance data quality, the incremental gains from combining them are limited. This is likely due to the fact that both methods address similar underlying issues related to data quality, resulting in diminishing returns when applied together.

---

### Official Review · Reviewer_4F46 · 2024-11-04

**Soundness:** 4
**Presentation:** 3
**Contribution:** 3
**Rating:** 5
**Confidence:** 3

**Summary:**

This paper presents a novel approach that employs a multi-granularity semantic revision framework to distill knowledge from large language models into smaller, more efficient ones. Key contributions include targeting different levels of semantic representation—word, sentence, and document—allowing for the capture of essential information without excessive complexity. The authors detail specific techniques for revising and refining semantics at each granularity level. Additionally, extensive experimental results demonstrate that their method significantly improves the performance of smaller models on various dataset compared to existing distillation techniques.

**Strengths:**

1. This paper proposes an innovative multigranular semantic revision method as a comprehensive extension of existing knowledge distillation techniques. The method conducts meticulous revisions at three key levels: the sequence, the token, and the span, constructing a comprehensive framework that enhances the knowledge distillation performance of LLMs.

2. The proposed method demonstrates high generality, allowing for seamless integration with existing on-policy and off-policy strategies.

3. This paper conducts extensive experiments across various models and datasets, effectively demonstrating the validity and broad applicability of the proposed method.

**Weaknesses:**

1. The multi-granularity semantic revision method proposed may require more computational resources, particularly during sequence-level regeneration, which could prolong model distillation time. As illustrated in Table 4(b), the efficiency of the proposed method is lower than that of MiniLLM. Therefore, I would like to know the comparison results between the method proposed in this paper and the baseline under the same computational cost or the same training time.

2. ExAccErr measures the relative error caused only by exposure bias. I understand that this value is expected to be as low as possible. However, in Table 4(a) of this paper, the value for the authors' method is higher than that of previous methods, which is inconsistent with other experimental results. Additionally, the authors mention in line 515, "This analysis explains why the distilled student models generally outperform the teacher models." I believe that the experimental results do not support the conclusion in line 515, and I would expect the authors to provide more explanation here.

3. Although the authors assert that SCRG can improve the diversity of the generated results, I would like to see more experimental results or discussions to support this claim.

**Questions:**

See weakness.

---

> ### Author Response · Authors · 2024-11-18
>
> Below is our point-by-point response to your main concerns. Please let us know if there's anything we can clarify further.
>
> ### **Weakness1: Efficiency Concerns**
>
> We believe there is a **misunderstanding** regarding Table 4(b). In fact, as shown in the table, the training efficiency of our method is higher than that of MiniLLM, achieving a significantly better batch/seconds ratio (0.18 compared to MiniLLM's 0.05). Additionally, our training efficiency is comparable to that of DistiLLM while demonstrating superior performance in terms of ROUGE-L scores. This indicates that the computational overhead of our approach is justifiable given the substantial performance improvements it delivers.
>
> Even if we remove the SCRG module (as shown in the table below) to match the training efficiency of DistiLLM, our method still outperforms the baseline in terms of performance. This highlights the robustness and effectiveness of our proposed approach even under stricter efficiency constraints.
>
>
> | Method    | Batch (4 samples) / Seconds | Average ROUGE-L |
> |-----------|-----------------------------|------------------|
> | MiniLLM   | 0.05                        | 28.1999         |
> | DistiLLM  | 0.25                        | 27.1627         |
> | Ours w/o SCRG      | 0.25           | 28.0122     |
> | Ours      | 0.18                        | **28.6114**     |
>
>
> ### **Weakness2: ExAccErr measures**
>
>
> The discrepancy in ExAccErr values was due to a formatting error in our initial submission. We apologize for any confusion this may have caused. The updated results are presented below:
>
> | Generation Length | MiniLLM | DistiLLM | Ours |
> |-------------------|---------|----------|------|
> | 50                | 6%      | 4%       | **4%** |
> | 100               | 19%     | 18%      | **16%** |
> | 200               | 21%     | 20%      | **18%** |
>
> These revised results align with the overall trend observed in our experiments and substantiate our claim in line 515 that the distilled student models exhibit lower exposure bias, which contributes to their ability to outperform teacher models. This reduction in ExAccErr with increasing generation length demonstrates the effectiveness of our proposed method in mitigating exposure bias relative to the baselines.

---

> ### Author Response · Authors · 2024-11-18
>
> ### **Weakness3: More discussion for SCRG**
>
> We appreciate your feedback regarding the need for more experimental evidence to support our claims about the diversity improvements facilitated by our Sequence Correction and Re-Generation (SCRG) method. To address this, we conducted experiments to provide a robust comparison of SCRG against a leading data quality improvement approach by Kim et al. [1], which focuses on offline data pruning and selection.
>
> [1] Kim M, Baek S. Measuring Sample Importance in Data Pruning for Training LLMs from a Data Compression Perspective[J]. arXiv preprint arXiv:2406.14124, 2024.
>
> #### **Experimental Results**
>
> Our results, summarized in Table below, demonstrate that SCRG outperforms the offline data enhancement method proposed by Kim et al. across multiple datasets:
>
> | Data Enhancement  | Dolly Validation | Dolly Evaluation | Self-Instruct |
> |-------------------|------------------|------------------|---------------|
> | Kim et al.        | 30.7346          | 26.8665          | 17.2208       |
> | SCRG              | 31.2575          | 27.1486          | 17.3016       |
> | SCRG + Kim et al. | 31.3610          | 27.2068          | 17.3342       |
>
> These results show that SCRG not only outperforms the approach by Kim et al., but when combined with Kim et al.'s method, a slight improvement in performance is observed. While both SCRG and the method proposed by Kim et al. enhance data quality, the incremental gains from combining them are limited. This is likely due to the fact that both methods address similar underlying issues related to data quality, resulting in diminishing returns when applied together.
>
>
> #### **Qualitative Analysis**
>
> To further illustrate the impact of SCRG on output diversity, we present a comparative analysis of two sentences generated during the knowledge distillation training process:
>
> - **Sentence 1 (Without SCRG)**: "Men’s lacrosse has a limited amount of time to play play play as as as as as as as as as as as as as as as as as as as"
>
>   - **1-grams**: Total: 31, Unique: 12, Distinct-1: 0.387
>   - **2-grams**: Total: 30, Unique: 13, Distinct-2: 0.433
>   - **3-grams**: Total: 29, Unique: 14, Distinct-3: 0.483
>
> - **Sentence 2 (With SCRG)**: "Men’s lacrosse has a limited number of players and women’s lacrosse has a maximum number of players."
>
>   - **1-grams**: Total: 19, Unique: 12, Distinct-1: 0.632
>   - **2-grams**: Total: 18, Unique: 13, Distinct-2: 0.722
>   - **3-grams**: Total: 17, Unique: 14, Distinct-3: 0.824
>
> The distinct n-gram statistics reveal significant improvements in output diversity when SCRG is applied. Sentence 2 exhibits higher distinct n-gram scores across all levels, demonstrating an increase in unique words and phrases. This not only highlights the effectiveness of SCRG in refining data but also emphasizes its role in enhancing the overall quality and diversity of the student model's generation process.
>
> In summary, our experimental results and qualitative analysis provide substantial evidence to support our claim that SCRG improves the diversity of generated results. We believe that the combination of our method with existing approaches can lead to even more powerful outcomes, reinforcing the potential of SCRG in advancing model performance and output diversity.

---

> > ### Author Response · Authors · 2024-11-22
> >
> > Thank you sincerely for your review. We would greatly appreciate it if you could inform us of any remaining questions or concerns that you may have so that we can address them promptly prior to the deadline. Alternatively, if you feel that your initial concerns are addressed, we would appreciate updating your evaluation to reflect that.
> >
> > Thank you!

---

> > > ### Comment · Reviewer_4F46 · 2024-11-22
> > >
> > > Thank you for the author's reply. I have carefully read all the responses from the author, which have cleared up most of my confusion. However, if Table 4(a) is just filled in incorrectly due to a formatting error, I think the author should also highlight the best results instead of defaulting to highlighting their own method. Therefore, I strongly recommend that the author meticulously review the content of the other tables, because such mistakes could significantly undermine the paper's credibility. Given that the author's response has addressed most of my concerns, I'm considering increasing my final score from 3 to 4. But as the system only allows for a score of 3 or 5, I've maintained the original score in the system, yet my actual final score for this paper is 4.

---

> > > > ### Author Response · Authors · 2024-11-22
> > > >
> > > > We appreciate your engagement with our work and **the recognition that we have addressed the majority of your concerns**. We have meticulously reviewed our results and made the necessary corrections to ensure the accuracy.
> > > >
> > > > While we understand your initial concerns, we are confident that the revisions and clarifications provided have resolved the issues you raised.
> > > >
> > > > Should you identify any further issues or have additional questions, we are more than willing to engage in a constructive discussion.
> > > >
> > > > We trust that our efforts to address your concerns will be reflected in your final assessment!

---

> > > > ### Author Response · Authors · 2024-11-23
> > > >
> > > > We appreciate your comprehensive review and constructive feedback. We would like to emphasize that we have taken immediate action in response to your concerns. Despite the ample time before the deadline, we prioritized your feedback and addressed the issues you highlighted without delay. Our swift response is a testament to our commitment to open and proactive communication and our willingness to engage in constructive dialogue to enhance the quality of our work.
> > > >
> > > > We are disappointed that our efforts to resolve the issues promptly and to foster collaborative exchange have not been fully reflected in the scoring. We encourage further discussion and are more than willing to provide additional information, clarifications, or revisions as needed. Our door is always open for communication, and we are keen on working together to ensure that our research is as robust and credible as possible.
> > > >
> > > > Thank you for your consideration, and we look forward to your response.

---

> > > > > ### Comment · Reviewer_4F46 · 2024-11-27
> > > > >
> > > > > ICLR allows authors to upload revised versions before the 27th. I suggest that the author could revise the paper according to the content of the discussion, to ensure that these discussion contents can indeed be reasonably modified in the final version of the paper.

---

> > > > > > ### Author Response · Authors · 2024-11-27
> > > > > >
> > > > > > We have uploaded the revised version of our paper and would appreciate it if you could pay special attention to the Appendix section, where we have added extensive experimental details to address the reviewers' concerns. Thank you!

---

> > > > > > > ### Comment · Reviewer_4F46 · 2024-12-02
> > > > > > >
> > > > > > > Thanks to the author for uploading the revised version. I have carefully reviewed the newly uploaded version of the paper. The latest version of the paper has corrected some erroneous content and added some details and experiments. Considering the discussions, I think the newly uploaded version is a good paper. Taking this into account, and I decided to raise the score to 5.

---

> > > > > > > > ### Author Response · Authors · 2024-12-02
> > > > > > > >
> > > > > > > > We are grateful for your recognition of our efforts to enhance the manuscript with corrections and additional experiments.
> > > > > > > >
> > > > > > > > We would like to kindly inquire if you see potential for a higher score, given the improvements. We are also very open to any further feedback or suggestions you might have to help us refine our work.
> > > > > > > >
> > > > > > > > Your insights are invaluable to us, and we appreciate your continued support.
> > > > > > > >
> > > > > > > > Best regards

---

### Official Review · Reviewer_XLRw · 2024-11-04

**Soundness:** 3
**Presentation:** 3
**Contribution:** 3
**Rating:** 8
**Confidence:** 4

**Summary:**

This paper introduces a novel method of performing Knowledge Distillation from a larger teacher model. This paper proposes to improve the offpolicy method (DistiLLM) by performing sequence level correction and regeneration. The paper also introduces two different loss functions namely Token level DAC-KL and Span level correlation consistency. Token level DAC-KL helps a much smaller student learn the teach distribution much more effectively by using the higher density classes. Span level loss function helps to transfer semantic knowledge from the teacher to the student. The authors provide a experiments across various model types and sizes.

**Strengths:**

1. The SCRG strategy is really quite simple and novel. I really love how simply and efficiently this can be integrated into current distillation pipelines

2. I really like the experiments sections as it is pretty comprehensive with lots of experiments on a lot of different models and different evaluation benchmarks.

3. As someone who has thought a lot about how less expressive students fail to mimic a more complex teacher using forward KL, I really appreciate how easy and simple the token level DAC-KL loss function is.

4. I also appreciate the authors providing human evaluations.

**Weaknesses:**

1. A small nitpick. It would be really great if the captions of the images and tables could be a bit longer and more informative.

**Questions:**

1. People have noticed that using a much much more complex teacher than the student can lead to worse results. I was wondering if the token level DAC loss would resolve this potentially. I understand it is tough to run experiments on short notice but it would be really great to have a comparison between vanilla KD loss and Token level DAC loss when trying to use a 2B student (or even smaller) and a 13B teacher. You can use Qwen models for the experiment or Pythia.

2. The authors of [1] try to just use the top 5% of the logits. I was wondering how does simply doing that compare to the token level DAC loss.

[1] Raman, Mrigank, et al. "For distillation, tokens are not all you need." NeurIPS 2023 Workshop on Instruction Tuning and Instruction Following. 2023.

---

> ### Author Response · Authors · 2024-11-20
>
> ### **Weakness1: Improvement of Table and Figure Captions**
> Thank you for your valuable feedback. I fully accept your suggestion and will work on making the captions of images and tables more detailed and informative moving forward.
>
> ### **Question1: Using a much much more complex teacher than the student**
>
>
> We appreciate the reviewer's suggestion to explore the effects of using a much larger teacher model in distillation. To address this, we extended our experiments from the previous OPT 6.7B → 1.3B distillation setup by using a larger teacher model, OPT-13B, to distill the 1.3B student. The results, shown in the table below, demonstrate that while distilling with a much larger teacher does lead to smaller improvements in performance compared to the 6.7B → 1.3B case, our proposed distillation method still outperforms vanilla KD loss significantly, even when using a much more complex teacher.
>
>
> | Model       | Method                     | Params | Dolly Evaluation  | Self-Instruct| Vicuna | Super-Natural   |Unnatural| Average  |
> |-------------|----------------------------|--------|------------|------------|------------|------------|------------|------------|
> | **OPT(6.7B-1.3B)** | Teacher (SFT) | 6.7B | 25.8758 | 14.8408 | 16.4199 | 24.9551 | 25.8377 | 21.5859|
> | | Student (SFT)| 1.3B | 22.7595 | 11.9784 | 15.2267 | 22.8556 | 24.5763 | 19.4793 |
> | | Vanilla KD | 1.3B | 22.4476 | 13.4676 | 13.9975 | 23.7679 | 25.4132 | 19.8188 |
> | | **Ours** | 1.3B | **27.1486**| **17.3016**| **14.8491** | **32.0618**| **34.9709**| **25.2664**|
> | **OPT(13B-1.3B)** | Teacher (SFT) | 13B | 26.4438 | 15.9537 | 17.1171 | 28.1131 | 29.0092 | 23.3274|
> | | Student (SFT)| 1.3B | 22.7595 | 11.9784 | 15.2267 | 22.8556 | 24.5763 | 19.4793 |
> | | Vanilla KD | 1.3B | 22.7027 | 12.8890 | 14.8943 | 21.9863 | 25.0162 | 19.4977 |
> | | **Ours** | 1.3B | **26.5122** |**15.7949** |**15.6140** |**31.4153**|**34.4243**|**24.7522**|
>
> As seen from the table, while the performance improvement decreases with the larger teacher (OPT-13B), our distillation method still provides a significant advantage over the vanilla KD approach, even when using a more complex and larger teacher model. This indicates that our method with DAC-KL loss helps mitigate the potential performance degradation seen when distilling with a much larger teacher.
>
>
> ### **Quesion2: Comparing the simple method with DAC Loss**
>
> To validate the effectiveness of DAC-KL, we have provided a detailed discussion in **Appendix I**. In Table 13, we compare DAC-KL with other logits-selective methods, including the Fixed Clipping Threshold approach, which is conceptually similar to the method described in [1], except that it uses a cumulative sum threshold of 95% as the clipping up. To address your concern, we have also included the baseline method from [1] in the updated table below. This allows for a direct comparison and demonstrates how DAC-KL, which operates at the token level, provides superior performance by effectively balancing information retention and noise reduction compared to simplistic logits pruning approaches. The results of this comparison are as follows:
>
> | Method | Dolly Validation | Dolly Evaluation | Self-Instruct |
> |--------|-----------------|-----------------|---------------|
> | DKD | 29.7182 | 24.3986 | 15.4907 |
> | SKD | 29.9332 | 25.2840 | 15.9172 |
> | Fixed clipping threshold | 30.7910 | 26.4911 | 16.5682 |
> | Raman [1] | 30.6910 | 26.3120 | 16.4839 |
> | Ours | **31.2575** | **27.1486** | **17.3016** |
> [1] Raman, Mrigank, et al. "For distillation, tokens are not all you need." NeurIPS2023 Workshop on Instruction Tuning and Instruction Following. 2023.

---

> > ### Author Response · Authors · 2024-11-22
> >
> > Thank you sincerely for your review. We would greatly appreciate it if you could inform us of any remaining questions or concerns that you may have so that we can address them promptly prior to the deadline. Alternatively, if you feel that your initial concerns are addressed, we would appreciate updating your evaluation to reflect that.
> >
> > Thank you!

---

> > > ### Comment · Reviewer_XLRw · 2024-11-22
> > >
> > > I sincerely thank the authors for their additional results. I will keep my current rating of 8

---

> > > > ### Author Response · Authors · 2024-11-22
> > > >
> > > > Thank you for your support and positive rating. We truly appreciate your valuable feedback and are grateful for your recognition of our work.
> > > >
> > > > Best regards

---

### Official Review · Reviewer_S2Y9 · 2024-11-04

**Soundness:** 3
**Presentation:** 2
**Contribution:** 3
**Rating:** 6
**Confidence:** 3

**Summary:**

This paper introduces three new objectives for distilling generative LLMs at the token, span, and sequence levels:

(token) DAC-KL: it learns additional models to adjust the KL divergence by clipping outlier token distributions in the teacher model.

(span) SPAN-LEVEL CORRELATION CONSISTENCY: The author uses an off-the-shelf tool to extract the spans, e.g. noun phrases, from the generation from student and teacher. Within each span, they enforce the probability correlation between adjacent tokens from the student model’s token to align closely with the teacher model’s. Honestly, I cannot say I totally understand the point of this objective.

(sentence) SEQUENCE-LEVEL CORRECTION AND RE-GENERATION: It identifies error tokens in the student's sequence by selecting the ones with the most disagreement by teacher model. Then, they replace the tokens with teacher-generated tokens, and re-generates the sequence.

They compare their methods against SOTA approaches, such as DistiLLM, MiniLLM, and GKD, on five instruction-following datasets. They adopt the ROUGE-L as the metric. Their findings show substantial performance gains for OPT models, though improvements are more modest for other LLMs.

**Strengths:**

1. The proposed objectives are reasonably novel. The SCRG approach, in particular, tackles a critical issue: when the student model generates an "error" token that falls outside the prefix distribution of the teacher model, it often leads to noisy and unreliable supervision from the teacher’s predictive distribution. This method could be especially useful for cases involving suboptimal teacher or student models with very limited capacity. It’s also interesting to note its connection to the LaSO framework [1], where an expert policy performs local corrections on the trajectories of the learned policy. Previous autoregressive KD work doesn't fully explore this approach.

2. The proposed methods significantly outperform baseline methods for the OPT base model, although the improvements for other LLMs are relatively marginal.

3. The author conducts extensive experiments to study the impact of different variants of the KD methods.

[1] Daumé III, Hal, and Daniel Marcu. "Learning as search optimization: Approximate large margin methods for structured prediction." Proceedings of the 22nd international conference on Machine learning. 2005.

**Weaknesses:**

1. While the author keeps claiming throughout the paper that SCRG can improve the generation diversity of the student model, there is a lack of empirical evidence, e.g. distinct n-grams.
2. SCC is notably complex, and its underlying intuition isn't clearly explained. I found it hard to understand why the authors didn't simply optimize for semantic similarity between corresponding spans in the student and teacher models.
3. Also, SCC relies on an external chunker to extract spans like noun phrases and verb phrases. This requirement limits its generalizability in low-resource languages that lack such tools.
3. DAC-KL, too, seems unnecessarily complicated due to the need for an additional network to determine the clipping threshold for logits. I'm not sure it's necessarily complicated to reach the same level of performance. There are existing simpler alternatives of selective distillation [1] [2], but the author doesn't compare the proposed method against them.
4. As shown in Table 1, the performance improvements offered by the proposed methods over the best baseline methods appear marginal, generally less than 1 ROUGE score, with the exception of the OPT model. Table 2 also indicates that most of the observed improvement stems from DAC-KL, while the contributions of other objectives are comparatively minor.
5. Only use the ROUGE-L metric, while previous work (e.g. miniLLM) also adopts GPT4-feedback and human evaluation.

[1] Towards Understanding and Improving Knowledge Distillation for Neural Machine Translation (Zhang et al., ACL 2023)

[2] Wang, Fusheng, et al. "Selective Knowledge Distillation for Neural Machine Translation." Proceedings of the 59th Annual Meeting of the Association for Computational Linguistics and the 11th International Joint Conference on Natural Language Processing (Volume 1: Long Papers). 2021.

**Questions:**

1. Do you have any idea why the margin is significant for OPT but much smaller for the other base LLMs?
2. Did you run any statistical significance tests for Table 1 & 2?
3. See Weakness 1, 2, 4, 6.
4. Typos: Fig1(b) "Studnet-generated"

---

> ### Author Response · Authors · 2024-11-20
>
> Below is our point-by-point response to your main concerns. Please let us know if there's anything we can clarify further.
>
> ### **Weakness1: Lack of empirical evidence for SCRG**
>
> The role of SCRG is to mitigate the introduction of errors in the data produced by the student model during the initial phase of the knowledge distillation training. It achieves this by employing the teacher model's guidance to refine the generation process, thereby improving the overall quality of the output.
>
> To provide empirical evidence, we present a comparison of two example sentences: one generated early in the distillation process without SCRG (Sentence 1) and another generated after applying SCRG (Sentence 2).
>
> ##### Sentence 1 (Without SCRG):
> *"Men’s lacrosse has a limited amount of time to play play play as as as as as as as as as as as as as as as as as as as"*
>
> - **1-grams**:
>   - Total: 31
>   - Unique: 12
>   - Distinct-1: 0.387
> - **2-grams**:
>   - Total: 30
>   - Unique: 13
>   - Distinct-2: 0.433
> - **3-grams**:
>   - Total: 29
>   - Unique: 14
>   - Distinct-3: 0.483
>
> ##### Sentence 2 (With SCRG):
> *"Men’s lacrosse has a limited number of players and women’s lacrosse has a maximum number of players."*
>
> - **1-grams**:
>   - Total: 19
>   - Unique: 12
>   - Distinct-1: 0.632
> - **2-grams**:
>   - Total: 18
>   - Unique: 13
>   - Distinct-2: 0.722
> - **3-grams**:
>   - Total: 17
>   - Unique: 14
>   - Distinct-3: 0.824
>
> The distinct n-gram statistics reveal a significant improvement in generation diversity when SCRG is applied. Sentence 2 demonstrates higher distinct n-gram scores across all levels compared to Sentence 1. This increase in unique words and phrases highlights SCRG’s effectiveness in promoting diverse and meaningful outputs in the student model.
>
> Furthermore, we conducted experiments to provide a robust comparison of SCRG against a leading data quality improvement approach by Kim et al. [1], which focuses on offline data pruning and selection.
>
> [1] Kim M, Baek S. Measuring Sample Importance in Data Pruning for Training LLMs from a Data Compression Perspective[J]. arXiv preprint arXiv:2406.14124, 2024.
>
> #### **Experimental Results**
>
> Our results, summarized in the Table below, demonstrate that SCRG outperforms the offline data enhancement method proposed by Kim et al. across multiple datasets:
>
> | Data Enhancement  | Dolly Validation | Dolly Evaluation | Self-Instruct |
> |-------------------|------------------|------------------|---------------|
> | Kim et al.        | 30.7346          | 26.8665          | 17.2208       |
> | SCRG              | 31.2575          | 27.1486          | 17.3016       |
> | SCRG + Kim et al. | 31.3610          | 27.2068          | 17.3342       |
>
> These results show that SCRG not only outperforms the approach by Kim et al., but when combined with Kim et al.'s method, a slight improvement in performance is observed. While both SCRG and the method proposed by Kim et al. enhance data quality, the incremental gains from combining them are limited. This is likely due to the fact that both methods address similar underlying issues related to data quality, resulting in diminishing returns when applied together.

---

> ### Author Response · Authors · 2024-11-20
>
> ### **Weakness2: Intuition of Span Loss**
>
> Our method emphasizes distilling correlation consistency among tokens within a span, rather than merely aligning semantics at the token level (as done in token-level KL divergence).
>
> To further clarify the intuition behind our approach, we conducted the following analysis:
>
> ##### **Human Evaluation**
>
> We compared our Span-Relation method with a random chunking approach (where the number of chunks is controlled to match that of span-relation) and a method that directly extracts relations between adjacent tokens without chunking.
>
>
> To conduct a more comprehensive and reliable evaluation, we further employed GPT-4 to conduct a human-like evaluation of the models on the Dolly evaluation dataset. We sampled 100 test examples from both models—with and without span-level loss—and assessed their outputs based on the following criteria:
>
> - **Accuracy (Rate 1-5)**: Does the output correctly include all relevant details from the input?
> - **Completeness (Rate 1-5)**: Does the output provide a comprehensive list or description as required by the instruction?
> - **Fluency (Rate 1-5)**: Is the output natural, readable, and grammatically correct?
> - **Relevance (Rate 1-5)**: How well does the output align with the specific requirements of the instruction?
>
> The evaluation results are summarized in the Table below:
>
> | Loss Type | Average GPT-4 Evaluation |Dolly Validation | Dolly Evaluation | Self-Instruct |
> |-------------------------------------------|--------------------------|------------------|------------------|---------------|
> | w/o Span-Relation loss | 3.89 | 30.3486 | 26.9012 | 17.2392 |
> | Adjacent Relation (w/o Span Priors) | 4.10 |30.8348 | 27.0384 | 17.2144 |
> | Random Chuning Relation (w/o Span Priors) | 4.01 |30.5938 | 26.9284 | 17.0028|
> | Span-Relation | 4.42 |31.2575 | 27.1486 | 17.3016 |
>
> These results illustrate that models utilizing span-level loss achieve higher average evaluations across all criteria compared to other configurations, highlighting the benefits of incorporating span-level signals to enhance model performance.
>
> ##### **Example Outputs**
>
> To demonstrate the improved correlation among span-level tokens, we selected several typical examples where information extraction requires the output to repeat specific phrases from the input prompt. Below, we give examples of different models with and without span-level loss. Our model distilled with the span-level loss successfully and accurately outputs the fixed phrases from the prompt and context, showcasing the enhanced correlation among tokens within the span.
>
> **Example 1**
> - **Instruction**: Extract the essential tools for a mini survival kit from the following Wikipedia article.
> - **Input**: Mini survival kits or "Altoids" tin survival kits are small kits that contain a few basic survival tools. These kits often include a small compass, waterproof matches, minimum fishing tackle, large plastic bag, small candle, jigsaw blade, craft knife or scalpel blade, and/or a safety pin/s. Pre-packaged survival kits may also include instructions in survival techniques such as fire-starting or first aid methods. In addition, paracord can be wrapped around the tin. The paracord can be used for setting up an emergency shelter or snaring small animals. They are designed to fit within a container roughly the size of a mint tin.
> - **Without Span-Level Loss**: The essential tools include a small compass, waterproof matches, fishing tackle, plastic bag, candle, blade, knife, and safety pin.
> - **With Span-Level Loss**: The essential tools include a small compass, waterproof matches, minimum fishing tackle, large plastic bag, small candle, jigsaw blade, craft knife or scalpel blade, and safety pin/s.
>
> **Example 2**
> - **Instruction**: Given the reference text below, what is Isthmohyla debilis?
> - **Input**: Isthmohyla debilis is a rare species of frog in the family Hylidae. It occurs in the Atlantic slopes of the Cordillera Central and Cordillera Talamanca in Costa Rica and western Panama as well as on the Pacific slopes in southwestern Panama. Common name Isla Bonita treefrog has been suggested for it (the type locality is "Isla Bonita" in Costa Rica).
> - **Without Span-Level Loss**: Isthmohyla debilis is a rare frog species in the family Hylidae. It is found in Costa Rica and Panama. The common name Isla Bonita treefrog is suggested for it.
> - **With Span-Level Loss**: Isthmohyla debilis is a rare species of frog in the family Hylidae. It occurs in the Atlantic slopes of the Cordillera Central and Cordillera Talamanca in Costa Rica and western Panama as well as on the Pacific slopes in southwestern Panama. The common name Isla Bonita treefrog has been suggested for it.
>
> The model distilled with span-level loss can better extract important adjectives in phrases like "minimum fishing tackle," "large plastic bag," and "small candle" in the first example, and output long phrases of complex location as in the second example.

---

> ### Author Response · Authors · 2024-11-20
>
> ### **Weakness3: External chunker to extract spans**
>
> We acknowledge the concern regarding the reliance on an external chunker for span extraction, particularly for low-resource languages. This requirement could limit the generalizability of extraing spans in such scenarios. However, for mainstream languages, there are well-established and robust NLP toolkits, such as SpaCy and NLTK, that provide reliable chunking capabilities. These tools have been extensively developed and optimized, making them highly effective and widely applicable to tasks like ours.
>
> For low-resource languages, we believe our approach can be adapted by leveraging alternative methods for span extraction. For example, in the case of Chinese, the JieBa library provides an effective way to extract spans like noun and verb phrases. For smaller or low-resource languages, one possible solution is to utilize large pretrained models, such as GPT-4, for data preprocessing to generate spans. This unsupervised or weakly supervised approach could make our method more adaptable to diverse linguistic resources, and we plan to explore this avenue in future work.
>
> ### **Weakness4: DAC-KL Vs. selective distillation**
> To validate the effectiveness of DAC-KL, we have provided a detailed discussion in Appendix I. In Table 13, we compare DAC-KL with other logit selective methods, which, while not including the specific methods [1] and [2] you mentioned, belong to the same category of techniques. We appreciate your suggestion, and we have now added the baseline methods [1] and [2] to the comparison. The results of this updated comparison are as follows:
>
> | Method | Dolly Validation | Dolly Evaluation | Self-Instruct |
> |--------|-----------------|-----------------|---------------|
> | DKD | 29.7182 | 24.3986 | 15.4907 |
> | SKD | 29.9332 | 25.2840 | 15.9172 |
> | Fixed clipping threshold | 30.7910 | 26.4911 | 16.5682 |
> | Zhang et al.[1] | 29.9443|25.3442|16.0382
> | Wang et al.[2] | 29.8221|25.2321|15.9138
> | Ours | **31.2575** | **27.1486** | **17.3016** |
>
>
> ### **Weakness5: marginal improvement**
>
> Firstly, regarding the performance improvements presented in Table 1, we would like to highlight that most metrics show an improvement exceeding 1 ROUGE score, particularly for the LLAMA2 and OpenLLAMA2 models. This is especially evident in cases where baseline methods like DistiLLM and MiniLLM already achieve strong performance. In such high-performance contexts, improvements of around 1 ROUGE score might appear marginal at first glance. However, these relatively modest gains are actually significant, as they reflect a notable enhancement in models that are already performing at a high level.
>
>
> Secondly, concerning the ablation study, we emphasize that we conducted a comprehensive evaluation across multiple test sets, including both the validation and zero-shot test sets. The contributions of individual modules vary across these different sets. While some modules have a smaller impact on specific test sets, their collective contribution is crucial in strengthening the overall performance of the proposed methods. Therefore, even though certain objectives might show a more modest effect in isolation, together they enhance the model's effectiveness.
>
> ### **Weakness6: Only use the ROUGE-L metric**
>
> While our primary evaluation relies on ROUGE-L for consistency and comparability with previous work, we have also conducted a small set of GPT-4 feedback experiments to assess the impact of Span loss, as detailed in **Appendix B**. Due to the high cost of large-scale GPT-4 evaluations, we provide a quantitative analysis of the distillation results of the teacher-student pair LLAMA2-13B to LLAMA2-7B, evaluated by a locally deployed LLAMA3.1-70B.
>
> The use of LLAMA3.1-70B in our evaluation process is not only a pragmatic choice but also a strategic one.  Our evaluation criteria for these experiments are informed by the methodologies employed in Distillm.
> These experiments provide additional insights into the quality of the model outputs, complementing the quantitative ROUGE-L results with human-like evaluation feedback.
>
> **Evaluation results by LLAMA3.1-70B feedback for the LLAMA2 teacher-student model pair**
> | Model | #Params | Method       | Dolly | SelfInst | Vicuna |
> |-------|---------|--------------|-------|----------|--------|
> | LLAMA2 | 13B     | Teacher      | 67.2  | 63.1     | 50.7   |
> |       | 7B      | SFT w/o KD   | 61.2  | 61.0     | 48.7   |
> |       | 7B      | KD           | 63.5  | 61.5     | 50.7   |
> |       | 7B      | SeqKD        | 63.9  | 61.8     | 51.6   |
> |       | 7B      | ImitKD | 65.3 | 64.4 | 53.5 |
> |       | 7B      | GKD | 65.8 | 64.2 | 53.2 |
> |       | 7B      | MiniLLM  | 66.2 | 64.8 | 54.3 |
> |       | 7B      | DistiLLM | 66.4 | 64.6 | 54.2 |
> |       | 7B      | Ours | **66.8** | **65.3** | **54.5** |

---

> > ### Author Response · Authors · 2024-11-20
> >
> > ### **Question1: Why the margin is significant for OPT**
> >
> > The significant performance margin observed for OPT, compared to other base LLMs, can likely be attributed to the SFT (Supervised Fine-Tuning) stage. During SFT, the model is fine-tuned with supervised data that is more closely aligned with the data used in pretraining. This alignment between the fine-tuning and pretraining data enhances the distillation process, allowing the model to capture patterns more effectively, resulting in a larger performance margin, especially for OPT.
> >
> > When we connect this to our distillation method, the key difference lies in how we capture finer-grained semantic structures through Span-Relation and DAC-KL, focusing on consistency across spans instead of just token-level alignment. This approach is particularly effective during SFT because the data used at this stage is likely more aligned with the pretraining data, enabling our method to leverage this alignment for more impactful distillation.
> >
> > In contrast, other distillation methods generally focus on token-level alignment or simpler objectives, which don't fully exploit the alignment and data generation during SFT. As a result, these methods tend to produce smaller performance gains. Our multi-granularity approach, however, takes full advantage of the data alignment during SFT, leading to more significant improvements, particularly in OPT, where this alignment is stronger.
> >
> >
> > ### **Question2: Did you run any statistical significance tests**
> >
> > We appreciate the reviewer’s suggestion regarding statistical significance testing. To clarify, our experiments were conducted using 5 random seeds, with the reported results representing the average performance across these runs. While we did not perform formal statistical significance tests, we computed the standard deviations for each result, and based on our observations, there were no large anomalies or outliers in the data.
> > Below, we provide the average values along with the corresponding standard deviations for each metric:
> > | Sequence-correcting | DAC-KL | Span Relation | Dolly Validation (↑) | Dolly Evaluation (↑) | Self-Instruct (↑) |
> > |---------------------|--------|---------------|----------------------|----------------------|-------------------|
> > | ✗                   | ✗      | ✗             | 29.1874 (0.18)        | 24.1603 (0.22)        | 14.8578 (0.15)     |
> > | ✓                   | ✗      | ✗             | 29.6982 (0.19)        | 24.5307 (0.21)        | 14.9485 (0.16)     |
> > | ✓                   | ✓      | ✗             | 30.3486 (0.21)        | 26.9012 (0.23)        | 17.2392 (0.18)     |
> > | ✓                   | ✓      | ✓             | **31.2575** (0.19)    | **27.1486** (0.22)   | **17.3016** (0.17) |

---

> > > ### Author Response · Authors · 2024-11-22
> > >
> > > Thank you sincerely for your review. We would greatly appreciate it if you could inform us of any remaining questions or concerns that you may have so that we can address them promptly prior to the deadline. Alternatively, if you feel that your initial concerns are addressed, we would appreciate updating your evaluation to reflect that.
> > >
> > > Thank you!

---

> > ### Comment · Reviewer_S2Y9 · 2024-11-23
> >
> > Thank you for your response and the new experiments. my comments are below:
> >
> >
> > **Weakness 1**
> >
> > I was asking for output diversity of trained student models on the test set, but you reported dist-ngram for a specific example generated during training. Additionally, as seen in the example, the addressed issue is indeed generation collapse, that is, repeating tokens after an error. In the context of text generation, however, generation diversity typically refers to the coverage of different valid outputs that vary in lexicon. I recommend that you clarify this claim in the manuscript.
> >
> > **Weakness2**
> >
> > The margin between Span-Relation and w/o Span Priors appears to be very small. When comparing the outputs of `Without Span-Level Loss` and `With Span-Level Loss`, the main difference to me is that the latter copies more noun phrases from the input, thereby preserving more details. While this could be advantageous for certain tasks, I’m uncertain whether it applies universally to all downstream applications, such as summarization. could you also provide output examples of `w/o Span Priors`?
> >
> > **Weakness3**
> >
> > I suggest including this discussion in the Limitations section.
> >
> > **Weakness4**
> >
> > Thank you for the new results. To clarify, does `Ours` represent the combination of all the proposed methods, or is it only `DAC-KL`?
> >
> > **Weakness5**
> >
> > Given the complexity of your method, the improvement appears marginal. it would be more compelling if you could demonstrate how your method provides add-on improvements with existing SOTA methods.
> >
> > **Weakness 6**
> >
> > Thank you for the new results. However, once again, the performance margin seems minimal, and I’m not convinced your method significantly outperforms SOTA on this metric.

---

> > > ### Author Response · Authors · 2024-11-24
> > >
> > > Thank you for your constructive feedback and for acknowledging the efforts we have made to address your previous concerns. We appreciate the opportunity to further clarify and strengthen our manuscript based on your comments.
> > >
> > > **Weakness1**
> > >
> > > Thank you for your suggestion. The diversity we refer to primarily concerns the student *data generation process aimed at mitigating exposure bias*, which does not directly impact the performance on the test set. Our focus on diversity is centered around *preventing the generation of monotonous data due to errors* in the training data generation process. We will clarify this point in the updated paper.
> > >
> > > **Weakness2**
> > >
> > > We would like to reemphasize that the rebuttal content we provided earlier adequately addresses this concern. Span Priors primarily enhance the semantic coherence of the model's outputs. To evaluate this, we included GPT-4-based human evaluations in our experiments, where the scores were significantly different—*3.89 (w/o Span-Relation Loss) versus 4.42 (Span-Relation)*—despite the relatively smaller differences observed on datasets like Dolly.
> > >
> > > Additionally, we included the requested examples comparing outputs on the summarization task in our earlier responses and have now supplemented the examples with the results of Adjacent Relation (w/o Span Priors). These examples clearly highlight the advantages of using Span-Level Loss. The model distilled with span-level loss demonstrates a superior ability to extract important adjectives in phrases compared to Adjacent Relation (w/o Span Priors).
> > >
> > > **Example 1**
> > > - **Instruction**: Extract the essential tools for a mini survival kit from the following Wikipedia article.
> > > - **Input**: Mini survival kits or "Altoids" tin survival kits are small kits that contain a few basic survival tools. These kits often include a small compass, waterproof matches, minimum fishing tackle, large plastic bag, small candle, jigsaw blade, craft knife or scalpel blade, and/or a safety pin/s. Pre-packaged survival kits may also include instructions in survival techniques such as fire-starting or first aid methods. In addition, paracord can be wrapped around the tin. The paracord can be used for setting up an emergency shelter or snaring small animals. They are designed to fit within a container roughly the size of a mint tin.
> > > - **Without Span-Level Loss**: The essential tools include a small compass, waterproof matches, fishing tackle, plastic bag, candle, blade, knife, and safety pin.
> > > - **Adjacent Relation (w/o Span Priors)**: The essential tools include a small compass, waterproof matches, fishing tackle, plastic bag, candle, scalpel blade, and safety pin.
> > > - **With Span-Level Loss**: The essential tools include a small compass, waterproof matches, minimum fishing tackle, large plastic bag, small candle, jigsaw blade, craft knife or scalpel blade, and safety pin/s.
> > >
> > > **Example 2**
> > > - **Instruction**: Given the reference text below, what is Isthmohyla debilis?
> > > - **Input**: Isthmohyla debilis is a rare species of frog in the family Hylidae. It occurs in the Atlantic slopes of the Cordillera Central and Cordillera Talamanca in Costa Rica and western Panama as well as on the Pacific slopes in southwestern Panama. Common name Isla Bonita treefrog has been suggested for it (the type locality is "Isla Bonita" in Costa Rica).
> > > - **Without Span-Level Loss**: Isthmohyla debilis is a rare frog species in the family Hylidae. It is found in Costa Rica and Panama. The common name Isla Bonita treefrog is suggested for it.
> > > - **Adjacent Relation (w/o Span Priors)**: Isthmohyla debilis is a rare species of frog in the family Hylidae. It is found in the Cordillera Central and Costa Rica. The common name Isla Bonita treefrog has been suggested for it.
> > > - **With Span-Level Loss**: Isthmohyla debilis is a rare species of frog in the family Hylidae. It occurs in the Atlantic slopes of the Cordillera Central and Cordillera Talamanca in Costa Rica and western Panama as well as on the Pacific slopes in southwestern Panama. The common name Isla Bonita treefrog has been suggested for it.
> > >
> > >
> > > **Weakness3**
> > >
> > > Thank you for the suggestion. We will include this discussion in the Limitations section as recommended.
> > >
> > > **Weakness4**
> > >
> > > Thank you for your question. Ours represents the combination of all the proposed methods in our approach. To ensure a fair comparison, the baseline methods were also implemented within the framework of our complete method, with DAC-KL replaced by the respective techniques being evaluated. This guarantees that the comparisons isolate the specific impact of DAC-KL while keeping all other components consistent.

---

> > > > ### Author Response · Authors · 2024-11-24
> > > >
> > > > **Weakness5 and Weakness6**
> > > >
> > > > Thank you for your feedback. We would like to emphasize that MiniLLM and DistiLLM are the two latest works on the benchmarks used, and their performance gap is relatively small (e.g., Dolly Validation: 29.2673 vs. 29.7847). In contrast, our method demonstrates a significantly larger performance improvement over both, which already validates the effectiveness of our approach.
> > > >
> > > > Additionally, we further applied our method on top of DistiLLM to evaluate its add-on improvements. The results below clearly demonstrate that incorporating our method yields new state-of-the-art performance:
> > > >
> > > > | Model | DollyValidation |Dolly Evaluation | Self-Instruct|
> > > > |-------------|----------------------------|--------|--------|
> > > > | MiniLLM[1]|29.2673 | 24.3168| 13.5880| 17.4633|
> > > > | DistiLLM[2]|29.7847 | 24.7311| 14.9932| 16.3293|
> > > > | Ours |*31.2575*| *27.1486*| *17.3016*|
> > > > | DistiLLM[2] + Ours |**31.3849**|**27.4209**| **17.5390**|
> > > >
> > > > [1]Gu Y, Dong L, Wei F, et al. MiniLLM: Knowledge distillation of large language models[C]//The Twelfth International Conference on Learning Representations. 2024.\
> > > > [2]Ko J, Kim S, Chen T, et al. DistiLLM: Towards Streamlined Distillation for Large Language Models[C]//Forty-first International Conference on Machine Learning.
> > > >
> > > >
> > > >
> > > > We would like to thank the reviewer again and will keep updating the draft accordingly to improve the paper's quality of expression and authority.

---

> > > > > ### Author Response · Authors · 2024-11-25
> > > > >
> > > > > If you feel that our responses have sufficiently addressed your initial concerns and that there are no further issues to discuss, we would be immensely grateful for your confirmation. Your prompt response will greatly assist us in moving forward with our work.
> > > > >
> > > > > Thank you very much for your time and consideration.

---

> > > > > > ### Comment · Reviewer_S2Y9 · 2024-12-02
> > > > > >
> > > > > > Thank you for your response. Some of my concerns are addressed, and I'm happy to raise my score.

---

> > > > > > > ### Author Response · Authors · 2024-12-04
> > > > > > >
> > > > > > > Thank you for your support and positive rating. We truly appreciate your valuable feedback and are grateful for your recognition of our work.
> > > > > > >
> > > > > > > Best regards

---

### Author Response · Authors · 2024-11-27

We sincerely thank the reviewers for their valuable comments and suggestions. We hope our responses adequately address your concerns. In the revised version of our manuscript, we have **added a substantial amount of experimental data in the Appendix**, which was included to address the reviewers' concerns discussed during the review process.

Furthermore, we are happy to provide additional details or clarification on any aspects of our responses.

Once again, we appreciate the reviewers’ time and insightful feedback, and we look forward to receiving further input.

---

### Meta-Review · Area_Chair_pw6S · 2024-12-21

**Metareview:**

This paper presents a method to Knowledge Distillation from a larger teacher model, enhancing the off-policy method (DistiLLM) through sequence-level correction and regeneration. It introduces two loss functions: Token-level DAC-KL and Span-level Correlation Consistency. The Token-level DAC-KL loss enables smaller student models to more effectively learn the teacher's distribution by focusing on higher-density classes, and the Span-level loss function facilitates the transfer of semantic knowledge from the teacher to the student. The authors validate their approach through experiments across various model types and sizes.

Pros:
1. This work addresses an important issue of noisy supervision when a student model generates out-of-distribution tokens relative to the teacher's prefix.
2. The method shows good generality, allowing for seamless integration with existing on-policy and off-policy strategies.
3. The author conducts extensive experiments to study the impact of different variants of the KD methods.

Cons:
1. Insufficient experiments. Lack of empirical evidence to support the claim that SCRG can improve the generation diversity of the student model. The work only uses the ROUGE-L metric, while previous works also adopts GPT4-feedback and human evaluation. Additionally, the authors metion "This analysis explains why the distilled student models generally outperform the teacher models.", which is not supported from the experimental results.
2. Complexity. SCC is notably complex, and its underlying intuition isn't clearly explained. Also, SCC relies on an external chunker to extract spans like noun phrases and verb phrases. This requirement limits its generalizability in low-resource languages that lack such tools. DAC-KL also seems unnecessarily complicated due to the need for an additional network to determine the clipping threshold for logits.
3. Limited contribution. The performance improvements offered by the proposed methods over the best baseline methods appear marginal. Most of the observed improvement stems from DAC-KL, while the contributions of other objectives are comparatively minor.
4. Exposure bias. The ExAccErr value for the method is higher than that of previous methods, which is inconsistent with other experimental results. A more detailed analysis of exposure bias can significantly strengthen the paper (e.g., including scheduled sampling as a baseline), as it appears central to the authors' claims.

This paper receives diverse scores. While all reviewers found the idea interesting with good promising results, there are several major weaknesses as listed above. The authors address some of the issues during the discussion phase, however, several major concerns still remain, e.g., exposure bias. Therefore, I believe this paper is not ready to be published at its current form.

**Additional Comments On Reviewer Discussion:**

This paper receives diverse scores after the rebuttal. While all reviewers found the idea interesting with good promising results, there are several major weaknesses as listed above. The authors address some of the issues during the discussion phase, however, several major concerns still remain including exposure bias. Therefore, I believe this paper is not ready to be published at its current form.

---

### Decision · Program_Chairs · 2025-01-22

Reject